# Data Center Cooling System Optimization Using Offline Reinforcement Learning

**Xianyuan Zhan**[1,2*†], **Xiangyu Zhu**[1*], **Peng Cheng**[1], **Xiao Hu**[1], **Ziteng He**[1], **Hanfei Geng**[1],
**Jichao Leng**[1], **Huiwen Zheng**[3], **Chenhui Liu**[3], **Tianshun Hong**[3], **Yan Liang**[3],
**Yunxin Liu**[1,2†], **Feng Zhao**[1†]
[1] Institute for AI Industry Research, Tsinghua University
[2] Shanghai Artificial Intelligence Laboratory    [3] Global Data Solutions Co., Ltd.
`{zhanxianyuan, liuyunxin}@air.tsinghua.edu.cn, fz@alum.mit.edu`

## Abstract

The recent advances in information technology and artificial intelligence have fueled a rapid expansion of the data center (DC) industry worldwide, accompanied by an immense appetite for electricity to power the DCs. In a typical DC, around 30∼40% of the energy is spent on the cooling system rather than on computer servers, posing a pressing need for developing new energy-saving optimization technologies for DC cooling systems. However, optimizing such real-world industrial systems faces numerous challenges, including but not limited to a lack of reliable simulation environments, limited historical data, and stringent safety and control robustness requirements. In this work, we present a novel physics-informed offline reinforcement learning (RL) framework for energy efficiency optimization of DC cooling systems. The proposed framework models the complex dynamical patterns and physical dependencies inside a server room using a purposely designed graph neural network architecture that is compliant with the fundamental time-reversal symmetry. Because of its well-behaved and generalizable state-action representations, the model enables sample-efficient and robust latent space offline policy learning using limited real-world operational data. Our framework has been successfully **deployed and verified** in a large-scale production DC for closed-loop control of its air-cooling units (ACUs). We conducted a total of 2000 hours of short and long-term experiments in the production DC environment. The results show that our method achieves 14∼21% energy savings in the DC cooling system, without any violation of the safety or operational constraints. We have also conducted a comprehensive evaluation of our approach in a real-world DC testbed environment. Our results have demonstrated the significant potential of offline RL in solving a broad range of data-limited, safety-critical real-world industrial control problems.

## 1 Introduction

With the surge of demands in information technology (IT) and artificial intelligence (AI) in recent decades, data centers (DCs) have quickly emerged as crucial infrastructures in modern society. Along with the rapid growth of the DC industry, comes immense energy and water consumption. In 2022, the global DC electricity consumption was estimated to be 240∼340 TWh, accounting for around 1∼1.3% of global electricity demand (International Energy Agency, 2023). It is forecasted that by 2026, the DC energy consumption in the US will rise to approximately 6% of the country's total power usage (International Energy Agency, 2024). To deal with the considerable amount of heat generated from servers and achieve temperature regulation, *cooling systems* typically account for about 30∼40% of total energy consumption in large-scale DCs (Van Heddeghem et al., 2014). Compared to server-side energy consumption that is primarily spent on computational tasks, reducing cooling energy consumption offers greater practical value for energy saving. How to improve the energy efficiency of DC's cooling systems while ensuring thermal safety requirements has become a critical problem for the DC industry, which has great economic and environmental impacts.

---

[*]Equal contribution.
[†]Corresponding authors.

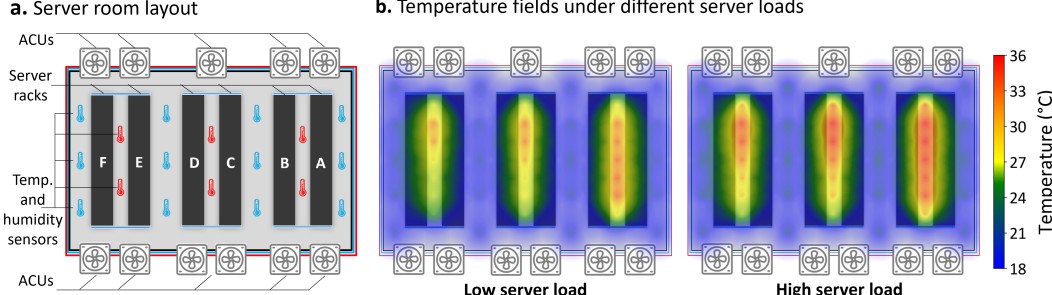

Figure 1: Illustration of the DC floor-level cooling system and temperature fields under different server loads.

In typical DCs, cold water generated from chillers and evaporative cooling towers is sent to multiple air-cooling units (ACUs) in the server rooms to provide cold air for servers. Properly controlling these ACUs in the server room is a challenging industrial control task. The difficulties arise from several aspects. First, frequently changing sever loads and physical locations of servers produce complex and dynamic temperature fields inside the server room (see Figure 1 as an illustration). Reaching the maximum degree of energy saving requires joint control of multiple ACUs in a way that is fully load-aware and capable of capturing complex thermal dynamics. Second, commercial DCs have very strict thermal safety and operational requirements, making it quite challenging to strike the right balance between energy efficiency and thermal safety. Lastly, due to the complex thermal dynamics of the cooling system, it becomes exceptionally hard to build high-fidelity and scalable simulators. Although there are many efforts (Chen et al., 2019; Ran et al., 2022a;b; Mahbod et al., 2022; Wang et al., 2022; Li et al., 2019; Chervonyi et al., 2022) that tried to build simulation environments based on techniques such as computational fluid dynamics (CFD) or multi-physics simulation, they suffer from nuanced system identification and calibration, while still having unavoidable large sim-to-real gaps. This makes the control policies learned using simulation-based online reinforcement learning (RL) methods hardly deployable in real-world DCs. Until now, most DCs still use conventional semi-automatic control methods, such as local Proportional-Integral-Derivative (PID) controllers for each ACU, which are manually tuned based on the expertise of human operators and operate conservatively to prevent overheating.

The recently emerged offline RL approach (Fujimoto et al., 2019; Zhan et al., 2022a) has provided an attractive data-driven and simulator-free solution to overcome the above-mentioned drawbacks. It offers a new possibility to learn policies directly from the historical operational data of DC cooling systems, and leverage highly expressive deep neural networks to overcome the low expressiveness and scalability issues in conventional PID (Durand-Estebe et al., 2013) and model predictive control (MPC) (Lazic et al., 2018; Mirhoseininejad et al., 2021; Ogawa et al., 2013) approaches. However, most existing offline RL algorithms require large amounts of training data with sufficient state-action space coverage to learn reasonable policies, otherwise will suffer from severe performance degradation (Li et al., 2022; Cheng et al., 2023). By contrast, although monitored by a large number of sensors, the historical operational data from real-world DC cooling systems are limited as compared to control complexity, and the data coverage is also quite narrow as they are generated from existing conventional controllers. This reality poses stringent requirements on the out-of-distribution (OOD) generalization and small-sample learning capability for a deployable offline RL model.

In this paper, we develop a physics-informed offline RL framework for energy-efficient DC cooling control. Specifically, we construct a special dynamics model to capture the complex thermal dynamics inside the server room, based on fundamental time-reversal symmetry (T-symmetry) compliance (Lamb & Roberts, 1998; Cheng et al., 2023) and graph neural network (GNN) (Kipf & Welling, 2017) architecture that embeds domain knowledge. Based on the well-behaved and generalizable latent representations provided by the model, we develop a sample-efficient offline RL algorithm, which learns and maximizes the value function in the latent space, while regularizing the agreement of policy-induced samples to both offline data distribution and T-symmetry consistency. The resulting algorithm enjoys great OOD generalization capability and is particularly effective given the limited real-world data availability.

Based on the proposed offline RL framework, we also developed a deployment-friendly system to facilitate real-world validation. Our system has been successfully **deployed and verified in a real-**

**world large-scale commercial data center**, achieving closed-loop control of its ACUs. Real-world validation experiments demonstrate that our system achieves **14-21% energy savings** in the DC cooling system without violating any safety or operational constraints during a total of **2000 hours** of short and long-term experiments. As production DC facilities do not tolerate any safety violations, we also build a **real-world small-scale DC testbed environment** (with 22 servers and an ACU) to fully evaluate and compare our approach against existing methods. Through comprehensive comparative experiments, our approach proves to be safe, effective, and robust as compared to other baseline methods. Last but not least, our approach has values not restricted to the scope of data center cooling, but also broadly applicable to other data-limited, safety-critical industrial control scenarios.

## 2 BACKGROUND AND RELATED WORK

**Data center cooling control optimization.** The cooling loop of typical DCs consists of water-side and air-side sub-systems. The former cools water with chillers and evaporative cooling towers, while the latter circulates the cold water to ACUs on the server floors. Through air-water heat exchange, the cooled air is blown out from the ACUs, regulating the air temperature in the server room. The generated warm water is then sent back to the chillers and cooling towers for re-cooling. In this study, we focus on the air-side cooling in the server room (also called *floor-level cooling* (Lazic et al., 2018)), where the primary goal is to optimize the fan speed (control the airflow) and valve opening (control the amount of cold water supply) in multiple ACUs, in order to achieve energy saving while meeting the room temperature requirements and ensuring thermal safety.

Traditional air-side cooling control methods include the local PID control (Durand-Estebe et al., 2013), the two-stage method (Lazic et al., 2018; Mirhoseininejad et al., 2021; Ogawa et al., 2013; Garcia-Gabin et al., 2018), and expert-based control (Gao & Jamidar, 2014). Specifically, local PID control relies on local sensor feedback to regulate the fan speed and valve opening of each individual ACU based on PID controllers, which is only applicable to small-scale control problems and unable to jointly optimize numerous ACUs. Two-stage methods first build a mechanism model and then apply optimization methods (such as MPC or linear quadratic control) to solve the cooling control problem based on the model. Both local PID and two-stage methods lack sufficient expressive power to capture complex state-action and dynamics patterns, and do not scale effectively with increasing problem size. Expert-based control leverages the experience and expertise of human operators to manage ACU cooling, requiring significant human labor and lacking transferability to different DC cooling systems. Recently, there have been many attempts to use online reinforcement learning (RL) to solve the DC cooling optimization problem (Chen et al., 2019; Ran et al., 2022b;a; Mahbod et al., 2022; Wang et al., 2022; Li et al., 2019; Chervonyi et al., 2022; An et al., 2023). However, these studies are restricted to simulation-based policy learning and validation. For a safety-critical industrial control scenario like DC cooling, it is nearly impossible to interact with real systems during policy training, and building a high-fidelity simulator can be very costly and impractical. This makes the previous online RL methods hardly have any success in real-world deployment.

**Offline reinforcement learning.** Offline RL aims to solve a sequential decision-making problem formulated by a Markov Decision Process (MDP), solely using a fixed offline dataset $\mathcal{D}$. The MDP is typically defined by a tuple $(\mathcal{S}, \mathcal{A}, T, r, \gamma)$ (Sutton & Barto, 2018), where $\mathcal{S}$ and $\mathcal{A}$ denote the state and action spaces, respectively. $T(s_{t+1}|s_t, a_t)$ denotes the transition dynamics. $r(s_t, a_t)$ denotes the reward function. $\gamma$ is the discount factor. Our goal is to learn an optimized policy $\pi^*(s)$ based on dataset $\mathcal{D}$ to maximize the discounted cumulative return, i.e., $R(\pi) = \mathbb{E}[\sum_{t=0}^{\infty} \gamma^t r(s_t, a_t)]$.

Under the offline setting, evaluating the RL value function in OOD regions can produce falsely optimistic values. Such exploitation errors can quickly build up during Bellman updates, eventually leading to severe value overestimation and misguiding policy learning. Hence most offline RL methods adopt various forms of data-related regularization schemes to stabilize policy learning, such as adding explicit or implicit behavioral constraints (Kumar et al., 2019; Fujimoto et al., 2019; Fujimoto & Gu, 2021; Li et al., 2022; 2023; Mao et al., 2024b), value regularization (Kumar et al., 2020; Xu et al., 2022c; Bai et al., 2021; Zhan et al., 2022b; Lyu et al., 2022; Niu et al., 2022), or adopt strict in-sample learning (Kostrikov et al., 2022; Xu et al., 2022a;b; Wang et al., 2024; Mao et al., 2024a). However, due to the exclusive use of strict data-related regularization, existing offline RL methods often suffer from over-conservatism and poor OOD generalization performance (Li et al., 2022; Cheng et al., 2023), which greatly restricts their usability in most data-limited real-

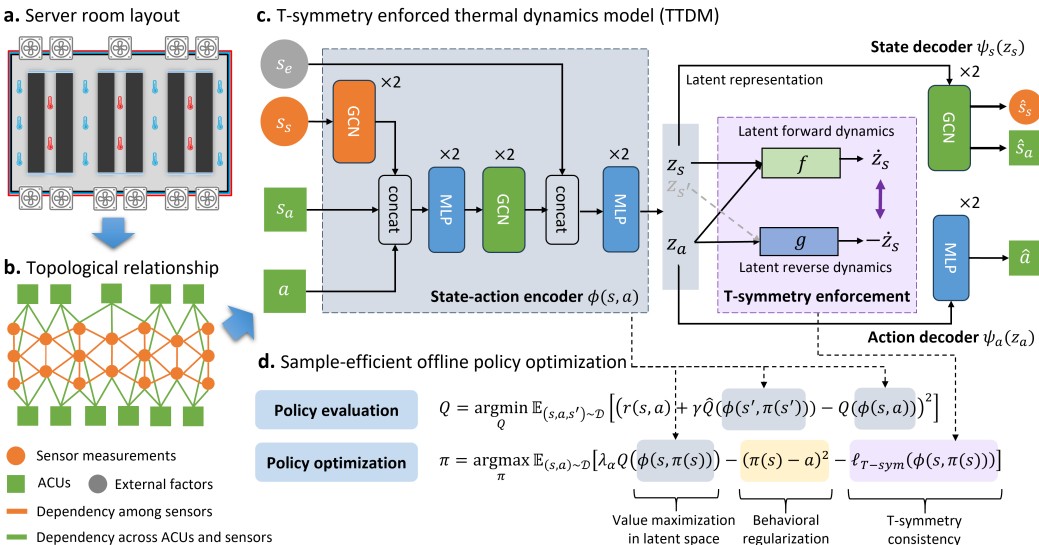

Figure 2: Illustration of the physics-informed offline RL framework for energy-efficient DC cooling control.

world control scenarios. The recently proposed T-symmetry regularized offline RL (TSRL) (Cheng et al., 2023) relaxes the restrictive data-related constraints by leveraging the fundamental time-reversal symmetry (i.e., the underlying laws of physics should not change under the time-reversal transformation: $t \to -t$), which significantly outperforms existing offline RL algorithms in terms of data efficiency and OOD generalization. Inspired by TSRL, we develop a new physics-informed offline RL framework and a system tailored to solving real-world complex, data-limited industrial control problems, such as DC cooling system optimization.

## 3 METHODOLOGY

In this study, we develop a physics-informed offline RL framework and a system to solve the DC cooling control optimization problem. We first mathematically formulate the problem into a standard MDP, with a specifically designed safety-aware reward function to ensure thermal-safe cooling control. The core component of our framework is a T-symmetry enforced Thermal Dynamics Model (TTDM), with a specifically designed GNN architecture to embed domain knowledge of spatial and control dependencies among sensors and ACUs. This model provides well-behaved and generalizable representations, enabling data-efficient and robust offline policy learning in the latent space. Based on the proposed offline RL framework, we also built an ACU control system that successfully deployed it in real-world DC environments. The overall framework of our method is illustrated in Figure 2.

### 3.1 PROBLEM FORMULATION AND REWARD DESIGN

As illustrated in Figure 1a, typical DC floor-level cooling systems involve several rows of server racks, flanked by two air handling rooms (AHRs) on each side, each containing 5 to 6 ACUs that blow cold air into the server rooms. The server racks are arranged with hot and cold aisles, utilizing hot (or cold) aisle containment. Temperature and humidity sensors are distributed throughout the aisles for overheating monitoring. The fan speed and valve opening can be controlled for each ACU to achieve desirable temperature regulation. However, commercial DCs have strict temperature requirements, improper control could cause cold aisle temperatures to exceed the safety threshold and negatively impact server operations. To solve this problem, we formulate it into a MDP with states, actions, and a reward function designed as follows:

**States.** The states of our problem $s = \{s_s, s_a, s_e\}$ contain three types of sensor inputs, including temperature and humidity sensor readings within the hot and cold aisles, and server rack temperature sensor readings, denoted as $s_s$; the working states of ACUs, such as leaving water temperature (LWT), leaving air temperature (LAT), and entering air temperature (EAT), denoted as $s_a$; and lastly, as

the entering water temperature (EWT) of each ACU and the server power consumption cannot be manipulated or controlled by the floor-level cooling system, they are considered as external factors, denoted as $s_e$. For real-world DC server rooms, the state vector $s$ typically has around 80 dimensions after feature engineering, collected at 2 to 5-minute intervals.

**Actions.** The action $a$ consists of the controllable variables for all ACUs in the server room, specifically the fan speed $f_m$ (control the airflow) and valve opening $o_m$ (control the amount of water) for each ACU $m$. For a server room with 11 ACUs, the complete action vector has 22 dimensions.

**Safety-aware reward function.** We design a reward function to balance energy saving and temperature regulation, taking into account both the operational parameters of the ACUs and the environmental factors within the cooling system. For an actual ACU $m$, high fan speed $f_m$ directly increases its power consumption, as the fan power consumption is proportional to the cube of the fan speed. On the other hand, a large valve opening $o_m$ could also marginally increase the energy consumption on the water-side sub-system. However, increasing fan speed and valve opening also improves the ACU's cooling effect, hence there is a complex trade-off. In terms of temperature safety constraints, our primary concern is whether the cold aisle temperature (CAT) $T_c^n$ monitored by the corresponding temperature sensor $n$ violates the safety threshold $\rho_T$. Additionally, for safety considerations and in consultation with on-site engineers' experience, we also regulate the LAT $T_l^m$ of ACU $m$ below a predefined threshold $\rho_L$. The resulting reward function is thus designed as:

$$r = r_0 - \beta_1 \sum_{m=1}^{M} f_m^3 - \beta_2 \sum_{n=1}^{N} \ln\left(1 + \exp\left(T_c^n - \rho_T\right)\right) - \beta_3 \sum_{m=1}^{M} o_m - \beta_4 \sum_{m=1}^{M} \ln\left(1 + \exp\left(T_l^m - \rho_L\right)\right)$$

$$(1)$$

where $r_0$ is a bias constant to keep the reward positive, $M$ is the number of ACUs, and $N$ is the total number of temperature sensors in the cold aisles. Positive coefficients $\beta_1, \beta_2, \beta_3, \beta_4$ are used to weight the respective terms in the reward function, balancing energy saving optimization and temperature regulation within the cooling system.

## 3.2 T-SYMMETRY ENFORCED THERMAL DYNAMICS MODEL

To extract robust and generalizable representations conducive to sample-efficient offline policy learning, we construct a special T-symmetry enforced thermal dynamics model (TTDM) to model and explain the fundamental thermal dynamics patterns inside the server room. More specifically, we start by abstracting the cooling system into two coupled graph structures with corresponding adjacency matrices as illustrated in Figure 2b. In this graph, each green node represents the ACU features $s_a$ and $a$, corresponding to its working states and controllable actions (fan speeds and valve openings), while each orange node represents sensor measurements $s_s$. As nearby sensor readings often have strong spatial correlation, while sensor readings themselves also have control dependencies on nearby ACUs, hence we connect these two types of nodes with orange and green edges to reflect the spatial and control dependencies respectively based on domain knowledge.

The detailed encoder-decoder architecture of TTDM is shown in Figure 2c. We design a state-action encoder $\phi(s, a)$ that contains a pair of GCN blocks (Kipf & Welling, 2017), to capture the spatial dependencies between sensor nodes (orange) and control dependencies across sensor nodes and ACU nodes (green). The external factors $s_e$ are integrated through a two-layer MLP to derive the final embedded representations. From the encoder $\phi(s, a)$, we can obtain the latent representations of the current state, action and next state $z_s, z_a, z_{s'}$ from data. To further enhance the reliability and generalizability of the learned representations, we introduce a pair of ODE latent forward dynamics $f(z_s, z_a) = \dot{z}_s$ and reverse dynamics $g(z_{s'}, z_a) = -\dot{z}_s$ to enforce the T-symmetry consistency ($f(z_s, z_a) = -g(z_{s'}, z_a)$). To learn this model, we design the following loss terms:

**Reconstruction loss.** As mentioned above, the state-action encoder $\phi(s, a) = (z_s, z_a)$ takes state-action pairs as input and outputs their corresponding latent representations. We then use a pair of state and action decoders $\psi_s(z_s)$ and $\psi_a(z_a)$ to ensure that the learned representations can be mapped back to the original data space:

$$\ell_{rec}(s, a) = \|s - \psi_s(z_s)\|_2^2 + \|a - \psi_a(z_a)\|_2^2 \qquad (2)$$

**Latent ODE forward and reverse dynamics.** We utilize the similar approach in Cheng et al. (2023) and Champion et al. (2019), embedding a discrete-time first-order ODE system to capture

the latent forward dynamics $f(z_s, z_a) = \dot{z}_s$, and the reverse dynamics $g(z_{s'}, z_a) = -\dot{z}_s$, where $\dot{z}_s = z_{s'} - z_s$. The reason that we model the latent dynamics as ODE systems is to encourage learning parsimonious models (Champion et al., 2019), which enables the model to capture more fundamental properties from the data, thereby helping avoid severe over-fitting that commonly occurs in small-sample learning situations and maximally promotes generalization. Note that based on the chain-rule, we can write $\dot{z}_s = \frac{dz_s}{dt} = \frac{\partial z_s}{\partial s} \cdot \frac{ds}{dt} = \nabla_s z_s \cdot \dot{s}$. Hence to enforce the ODE property, we can use the following loss to train $f$ and $g$:

$$\ell_{fwd}(s, a, s') = \|(\nabla_s z_s)\dot{s} - \dot{z}_s\|_2^2 = \left\|\frac{\partial \phi(s, a)}{\partial s}\dot{s} - f(\phi(s, a))\right\|_2^2 \tag{3}$$

$$\ell_{rvs}(s, a, s') = \|(\nabla_{s'} z_{s'})(-\dot{s}) - (-\dot{z}_s)\|_2^2 = \left\|\frac{\partial \phi(s', a)}{\partial s'}(-\dot{s}) - g(\phi(s', a))\right\|_2^2 \tag{4}$$

Moreover, we also require the state decoder $\psi_s(z_s)$ has the capability to decode $\dot{s}$ from $\dot{z}_s$ (i.e., $\psi_s(\dot{z}_s) = \dot{s}$), to ensure it is compatible with the ODE property. This implies the following loss:

$$\ell_{ds}(s, a, s') = \|\dot{s} - \psi_s(\dot{z}_s)\|_2^2 \tag{5}$$

**T-symmetry regularization.** To obtain a well-behaved latent representation derived from $\phi$, we enforce an adapted version of T-symmetry for the discrete-time MDP setting (Cheng et al., 2023), by constraining the two latent ODE dynamics to satisfy $f(z_s, z_a) = -g(z_{s'}, z_a)$. This leads to the following T-symmetry consistency loss:

$$\ell_{T\text{-}sym}(z_s, z_a) = \|f(z_s, z_a) + g(z_s + f(z_s, z_a), z_a)\|_2^2 \tag{6}$$

Note that in above loss term, we leverage the fact $z_{s'} = z_s + \dot{z}_s = z_s + f(z_s, z_a)$ and use $g(z_s + f(z_s, z_a), z_a)$ instead of $g(z_{s'}, z_a)$ to further couple the learning process of $f$ and $g$. We find this treatment can better regulate the learning process of the latent ODE forward and reverse dynamics in our empirical experiments.

**Final learning objective.** Finally, the complete loss function of TTDM is:

$$\mathcal{L}_{TTDM} = \sum_{(s, a, s') \in \mathcal{D}} [\ell_{rec} + \ell_{fwd} + \ell_{rvs} + \ell_{ds} + \ell_{T\text{-}sym}](s, a, s') \tag{7}$$

### 3.3 SAMPLE-EFFICIENT OFFLINE POLICY OPTIMIZATION

We construct a highly sample-efficient offline RL algorithm for energy-efficient DC cooling control by integrating the properties of the learned TTDM. The most notable benefit of leveraging TTDM in offline policy learning lies in the well-behaved compact data representations produced by its state-action encoder $\phi(s, a)$, which are both information-rich (capturing fundamental dynamics information) and robust (well-regularized and T-symmetry preserving). This can greatly enhance offline policy learning and generalization on OOD areas, crucial for the small-sample learning setting. Consequently, instead of learning the action-value function in the original data space as in typical RL algorithms, we learn our action-value function within the latent space (i.e., $Q(z_s, z_a)$). This provides more reliable value estimates even with limited offline data. Specifically, we update our $Q$-function using the following objective with the safety-aware reward function defined in Eq. (1):

$$Q = \underset{Q}{\arg\min}\, \mathbb{E}_{(s, a, s') \sim \mathcal{D}}\left[\left(r(s, a) + \gamma \hat{Q}\left(\phi\left(s', \pi\left(s'\right)\right)\right) - Q(\phi(s, a))\right)^2\right] \tag{8}$$

For policy optimization, we adopt a similar treatment as in TD3+BC (Fujimoto & Gu, 2021), where we maximize the value function $Q$ but in the latent space, and constrain the policy output actions closer to actions within the dataset. However, solely adding the regularization to offline behavioral data is insufficient to ensure reasonable generalization performance. Hence we further regularize the T-symmetry consistency of policy-induced samples $(s, \pi(s))$ using the T-symmetry consistency loss $\ell_{T\text{-}sym}$ as in Cheng et al. (2023). This enforces the policy to generate actions that are compliant with T-symmetry, even in OOD areas, thereby greatly enhancing the generalization performance and sample efficiency of policy learning. The final policy optimization objective is presented as follows:

$$\pi = \underset{\pi}{\arg\max}\, \mathbb{E}_{(s, a) \sim \mathcal{D}}\left[\lambda_\alpha Q(\phi(s, \pi(s))) - (\pi(s) - a)^2 - \ell_{T\text{-}sym}(\phi(s, \pi(s)))\right] \tag{9}$$

where we follow TD3+BC and use $\lambda_\alpha = \alpha / [\sum_{s_i, a_i} |Q(\phi(s, a))|/N]$ as the normalization term to balance the strength of value maximization and policy regularization ($N$ is the number of samples in a training batch). We tuned the scale parameter $\alpha$ in the range of $[2.5, 10]$ during our experiments.

Table 1: Comparison of conventional PID control and our approach under comparable server load settings on two server rooms of the commercial DC. "AEP" and "EC" denote average electric power and energy consumption, respectively. We use the offline RL policy to control 4, 6, and all the ACUs in each room. ACLF is the air-side cooling load factor, calculated as the ratio of energy consumption of ACUs to servers, the lower the better.

| Server Room A | PID | | Ours (4 ACUs) | | | Ours (6 ACUs) | Ours (all ACUs) |
|---|---|---|---|---|---|---|---|
| | May 5th 11:00 - 17:30 | May 6th 09:50 - 17:20 | May 7th 11:00 - 17:30 | May 8th 09:50 - 17:20 | May 9th 09:50 - 17:20 | Sep 23 11:00 - Sep 29 10:30 | Nov 11 16:30 - Nov 12 16:30 |
| Server AEP (kW) | 555.31 | 552.17 | 548.61 | 549.28 | 550.19 | 572.77 | 577.63 |
| Server EC (kWh) | 3610.15 | 4141.34 | 3566.65 | 4120.42 | 4127.38 | 82199.92 | 13864.55 |
| ACU AEP (kW) | 24.53 | 23.82 | 19.9 | 20.19 | 20.00 | 20.78 | 19.66 |
| ACU EC (kWh) | 159.42 | 178.73 | 129.24 | 151.44 | 149.97 | 2981.84 | 471.8 |
| ACLF (%) | 4.42 | 4.32 | 3.62 (↓18%) | 3.68 (↓15%) | 3.63 (↓16%) | 3.63 (↓16%) | 3.40 (↓21%) |
| Server Room B | PID | | Our (4 ACUs) | | | Ours (6 ACUs) | Ours (all ACUs) |
| | May 5th 11:00 - 17:30 | May 6th 09:50 - 17:20 | May 7th 11:00 - 17:30 | May 8th 09:50 - 17:20 | May 9th 09:50 - 17:20 | Sep 23 11:00 - Sep 29 10:30 | Oct 30 10:10 - Nov 1 17:30 |
| Server AEP (kW) | 617.18 | 602.04 | 593.28 | 610.57 | 611.34 | 576.52 | 619.55 |
| Server EC (kWh) | 4010.83 | 4520.42 | 3853.19 | 4579.69 | 4586.52 | 82746.24 | 34302.42 |
| ACU AEP (kW) | 37.2 | 36.38 | 30.58 | 31.66 | 31.76 | 29.15 | 30.06 |
| ACU EC (kWh) | 241.79 | 272.9 | 198.75 | 237.43 | 238.15 | 4183.44 | 1663.22 |
| ACLF (%) | 6.03 | 6.04 | 5.16 (↓14%) | 5.18 (↓14%) | 5.19 (↓14%) | 5.06 (↓16%) | 4.85 (↓20%) |

## 4 REAL-WORLD EXPERIMENTS

To validate our proposed physics-informed offline RL framework, we develop a deployment-friendly software system to support the close-loop control of ACUs using the learned policy. We successfully deployed our system and conducted a series of experiments (from January to December 2024) in a large-scale commercial data center in China, controlling up to 4, 6, and all (10 or 11) ACUs in two of its server rooms (referred as Room A and B in the later content). Our method has been operated effectively and safely for over 2000 hours in total. As conducting experiments in a production environment suffers lots of restrictions, to further validate our method, we also built a real-world small-scale DC testbed to conduct more comprehensive comparative experiments and model ablations. The testbed contains 22 servers and an ACU, and supports testing a wide range of server load settings. More information about the two real-world DC testing environments and the collected historical operational datasets can be found in Appendix B. Throughout this study, we train and validate our model on real-world data and environments, with completely no simulation involved.

### 4.1 VALIDATION ON REAL-WORLD DATA CENTER

**Comparison with conventional control.** We first compare our DC cooling optimization method with the default ACU PID controllers on two server rooms in the real-world commercial data center. As our experiments are conducted in the real production environment, we are only allowed by the DC operator to control 4 out of 11 ACUs in the room in the early stages of the experiment, once the effectiveness was validated, we proceeded with experiments controlling 6 ACUs and then all ACUs in a server room. To ensure a fair comparison, we select several time periods (lengths from 5.5 to 7.5 hours) that have similar server load patterns for comparison. Table 1 shows the results on energy consumption metrics, including the average electric power and total energy consumption of servers and the ACU cooling system. We use the *Air-side Cooling Load Factor* (ACLF) to analyze the floor-level cooling system's energy efficiency, which is widely adopted by the DC industry. It is calculated as the ratio of the ACU system's energy consumption to the servers' energy consumption during the test period. Lower ACLF indicates higher energy efficiency. In the tested two server rooms, our method improves the cooling system's energy efficiency by 14% to 21% compared to the default PID controllers. Throughout our experiment, we observed no thermal safety violations and regulated the cold aisle temperature (CAT) well below the required operational threshold.

**Control quality.** We also conducted consecutive 48-hour experiments to compare the control behaviors of our method and the PID controllers in Server Room B with fluctuating server loads. The results are presented in Figure 3, where we compare the same 4 controlled ACUs and the temperature variation patterns of the directly impacted hot and cold aisles. As shown in Figure 3a, during the periods controlled by the PID controllers and our method, the total server load fluctuated

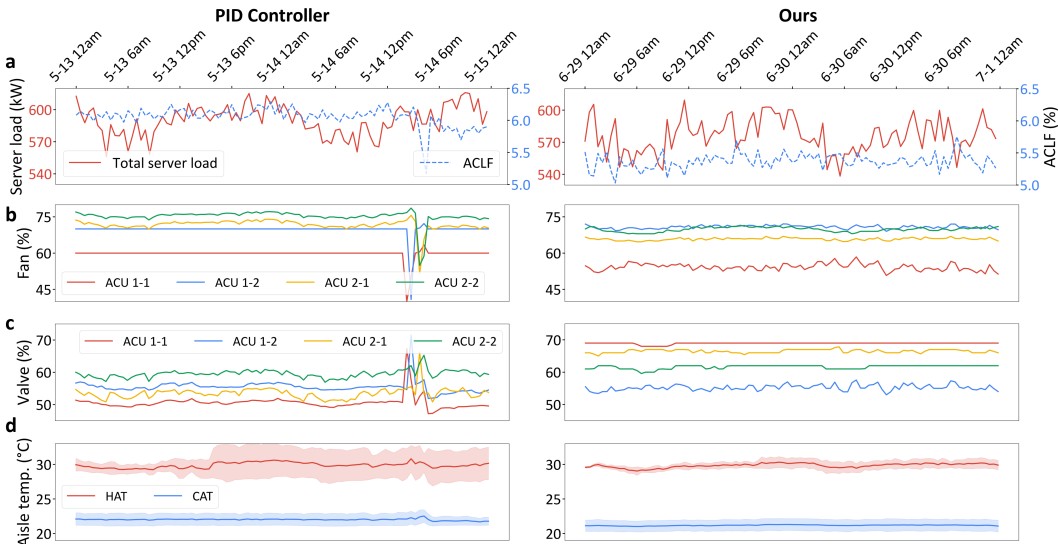

Figure 3: Comparisons of key system metrics and the controllable actions of our method and the PID controller over 2-day testing periods in Server Room B. Figures on the left show results from the PID-controlled period (May 13-15, 2024), and figures on the right are the results controlled by our method (June 29 - July 1, 2024).

at a similar level, but our method consistently achieved noticeably lower ACLF value than that of the PID controller, indicating higher energy efficiency. In Figures 3b and 3c, we compare the controllable actions (fan speeds and valve openings) of the 4 controlled ACUs during the test period. For fan speeds, several ACUs controlled by the default PID controller remained almost constant for a long time, whereas the fan speeds of the 4 ACUs controlled by our method were dynamically adjusted throughout the testing period. Notably, during the PID control phase, there was a short period having drastic adjustments in fan speed and valve opening, while such abnormal control behavior was not observed in our method. In terms of overall control behavior, our method tends to lower the fan speeds while slightly increasing the cold water valve openings, which helps reduce ACU energy consumption while maintaining the same level of cooling capacity. Figure 3d shows hot and cold aisle temperature variations during the testing periods. The solid curve and shaded area represent the mean and the mean±std envelop of multiple temperature sensor readings. Our method slightly decreased the cold aisle temperature, even with less ACU energy consumption (lower ACLF). Moreover, we find that our method achieves significantly better temperature regulation for the hot aisle, which results in much more concentrated temperature distributions as compared to the PID controller, indicating a more uniform and stable temperature field inside the hot aisle. More results that showcase the superior adaptability of our method under drastic server load fluctuations can also be found in Appendix C.1.

**Long-term control performance.** To verify the long-term robustness and energy-saving effectiveness of our method, we conducted two 14-day experiments by continuously running our offline RL policy and the PID controller on the 4 controllable ACUs in Server Room B. Our model was in operation from June 17 to July 1, 2024, while the PID controller was in operation from July 2-16, 2024. Figure 4 presents the results of energy efficiency and temperature conditions of the directly influenced cold and hot aisles (see Appendix B.1 for details). In Figure 4a, each point represents the average total server load within an hour and the corresponding calculated ACLF value. The ACLF values of our model are consistently lower than those of the PID controller across all server load conditions, with even lower ACLF values observed under higher server loads. This again demonstrates the load-awareness of our approach, which enjoys a greater level of energy saving with the increase of server loads, forming a sharp contrast to the almost constant ACLF level of the PID control. Figure 4b illustrates the temperature distribution in the most relevant hot aisle, where the PID controller resulted in a distribution clustering around 29°C and 31.5°C. By contrast, our method maintained a more concentrated temperature distribution around 30°C, leading to a more uniform temperature field inside the hot aisle during the testing period. Figure 4c shows the temperature distribution of the two most relevant cold aisles during the 14-day experiments, both methods regulated the cold aisle temperature below the operational threshold of 25°C. These results demonstrates the potential of our method for safe and stable long-term deployment in real-world data centers.

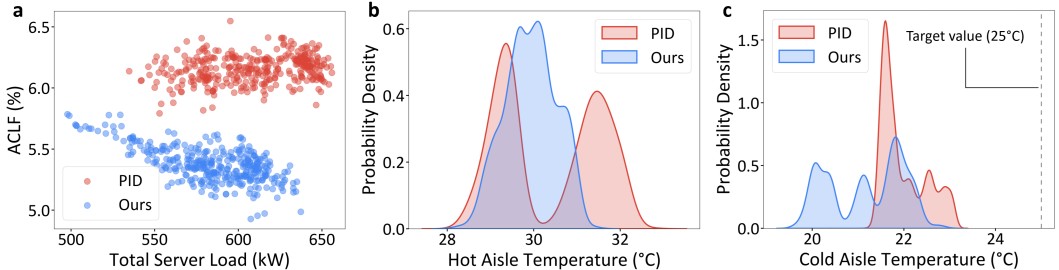

Figure 4: Results of the 14-day long-term experiments in Server Room B. **a**, ACLF values under different total server loads. **b**, **c**, Temperature distribution of the directly influenced hot and cold aisles.

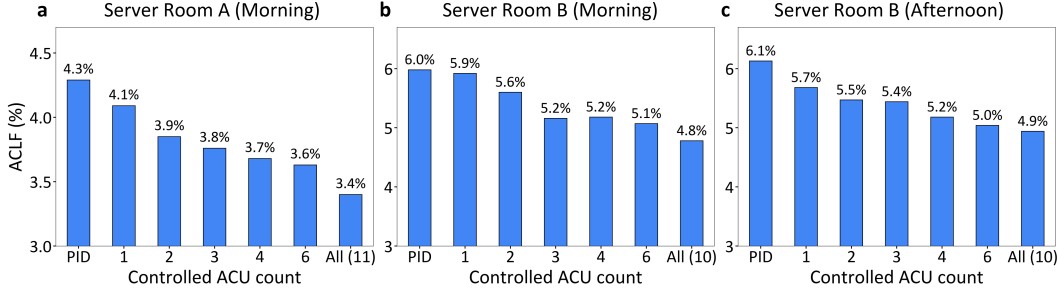

Figure 5: The energy-saving impact of controlling different numbers of ACUs through our approach.

**Impact of the number of controlled ACUs.** We also conducted additional experiments with our model controlling 1 to all ACUs to further investigate its energy-saving impact. The results are presented in Figure 5, which clearly show an increasing trend of energy efficiency with more ACUs controlled by our method. Figure 5a shows the experiment results conducted in seven morning periods (10:30 - 13:30) in Server Room A; Figure 5b,c on the right show the experiment results conducted in seven morning (10:30 - 13:30) and afternoon (14:30 - 17:30) periods in Server Room B. These promising results suggest that if more ACUs can be controlled by our method, it is very likely that we can achieve even higher energy efficiency.

## 4.2 EVALUATION AND ABLATION ON THE TESTBED

As testing in the production DC environment suffers lots of restrictions, to further validate our method, we conducted extensive exploratory experiments and model ablations in our testbed environment.

**Comparative evaluation against baseline methods.** We compare our method with competing baseline methods including conventional industrial control methods PID and MPC (Lazic et al., 2018), off-policy RL-based DC cooling optimization method CCA (Li et al., 2019), mainstream offline RL algorithms IQL (Kostrikov et al., 2022) and CQL (Kumar et al., 2020), and the state-of-the-art safe offline RL algorithm FISOR (Zheng et al., 2024) (see Appendix D.2 for detailed descriptions). For the comparative experiments, we tested three server load conditions: low, medium, and high loads, with average electric power of 4.9kW, 7.4kW, and 8.0kW, respectively. Each method controlled the ACU in closed-loop mode for 6 hours under the same experimental conditions, and we recorded the energy efficiency and thermal safety metrics, i.e., ACLF and CAT violations (proportion of time steps during the experiment that the CAT exceeds the pre-defined threshold). To make the task more challenging, we set a lower CAT threshold (22°C) as compared to the one used in the commercial DC to test the capability of the algorithm in balancing energy saving and temperature regulation. The results are reported in Figure 6. Due to the smaller scale of the testbed and significantly lower server load as compared to the real-world DC, the calculated ACLF values are higher than those observed in the real DC experiments. We observe some aggressive baseline methods (CCA and CQL) achieve lower energy consumption but perform poorly in terms of thermal safety, which is unacceptable. By contrast, our method achieved the highest energy efficiency under all load conditions, while ensuring no CAT violations throughout the experiments, outperforming all other baseline methods.

**Ablation study.** In addition, we conducted ablation experiments to validate the effectiveness of key designs in our method, including the GNN architecture and T-symmetry enforcement. Additional

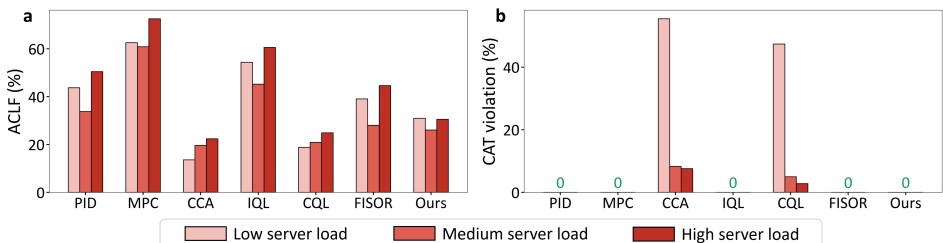

Figure 6: Comparative evaluation of our method against baseline methods on our real-world testbed.

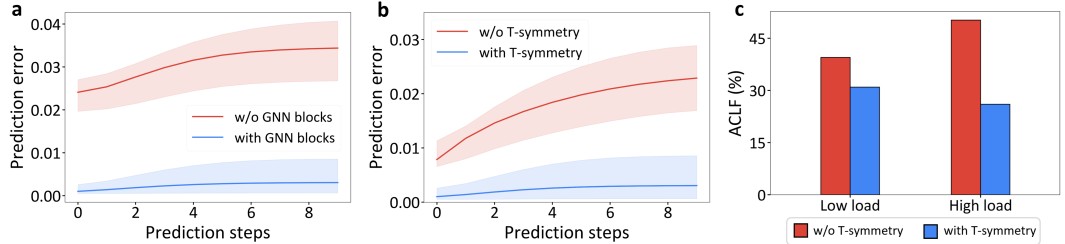

Figure 7: Ablation experiments on the impact of GNN blocks and T-symmetry enforcement in our method.

ablation results on the reward function design can be found in the Appendix C.2. In Figure 7a, b, we compare the multi-step prediction error of our proposed TTDM trained on the historical data of Server Room B with and without the GNN structure and T-symmetry enforcement. The prediction errors are measured in terms of mean square error (MSE) on the predicted future states. The results show that incorporating domain knowledge (spatial and control dependencies among sensors and ACUs) using GNN blocks significantly reduces TTDM's prediction error, especially when the number of prediction steps increases. We also obtain similar results when incorporating T-symmetry enforcement in TTDM, which demonstrates that both the GNN architecture and T-symmetry design can substantially improve the capacity and generalization of our thermal dynamics model, thereby providing better modeling and representation of the offline dataset. In Figure 7c, we compare the offline policy optimization results of our method with and without T-symmetry under low and high server load conditions on the testbed. Each experiment ran continuously for 6 hours. The results show that, the version of our offline RL framework with T-symmetry achieves much better energy efficiency improvements in both load conditions compared to the version without T-symmetry. This indicates that T-symmetry plays a crucial role in enhancing the generalization during policy learning, therefore resulting in more performant policy given limited real-world data.

## 5 CONCLUSION

In this study, we develop a physics-informed offline RL framework and a deployable system for energy-efficient DC cooling control. The core of our framework is a graph-structured and T-symmetry consistent thermal dynamics model, which provides well-behaved and generalizable representations, enabling highly sample-efficient offline policy learning in the latent space. Our system has been successfully deployed and validated in a real-world large-scale commercial data center and achieved closed-loop control of its ACUs. Our empirical results show that our proposed method can achieve 14∼21% energy savings in the real-world DC cooling system, and ran smoothly without any safety or operational constraints violation during long-term experiments. We also provide comprehensive comparative evaluations and ablations of our approach in a real-world small-scale DC testbed environment that is constructed specifically for this research. Our work demonstrates the huge potential of offline RL in solving a broad range of complex real-world industrial control problems, especially for those having limited historical data and impossible to build high-fidelity simulators. Lastly, we also urge the RL community to move away from current toy simulation-based RL benchmark environments and focus more on real-world control problems. The current simulation-based RL benchmarks have many unrealistic and biased dataset/task settings, which often provide misleading insights that mismatch with observations in real-world practices.

## ACKNOWLEDGMENTS

This work is supported by Carbon Neutrality and Energy System Transformation (CNEST) Program, and funding from Global Data Solutions Co., Ltd and Wuxi Research Institute of Applied Technologies, Tsinghua University under Grant 20242001120. We are especially grateful for all the support from Global Data Solutions Co., Ltd in our real-world DC experiments.

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

APPENDIX

## A    SYSTEM DEPLOYMENT

We have developed a full-function software system to facilitate the deployment and validation of our proposed physics-informed offline RL framework. We successfully deployed our system in a large-scale commercial data center for production environment performance validation and the small-scale DC testbed for more comprehensive model evaluation and ablation. The overall deployed system architecture is illustrated in Figure 8, which consists of two main phases: offline training and online deployment. In the offline training phase, the historical operational data of the floor-level cooling systems is exported from the DC log management system. The exported data undergoes automated data processing and feature engineering processes and is stored in a historical dataset. Subsequently, based on the processed offline dataset, we train the T-symmetry enforced thermal dynamics model, followed by a sample-efficient offline policy learning module to obtain the optimized floor-level cooling control policy. In the online deployment phase, the learned policy is deployed in a local policy server within the data center to provide control services. Real-time data from the cooling systems is retrieved by the management system API, processed, and stored in a real-time database. The system then forwards the real-time data to the policy server, which outputs optimized ACU control actions. These optimized control actions are directly written into the ACUs via the Modbus protocol for closed-loop control.

Our developed system is deployment-friendly and broadly applicable to various DC floor-level cooling systems with different configurations, exhibiting great flexibility and transferability. Moreover, as environmental and server load conditions in the data center frequently change over time, the completely data-driven design of our system offers extra advantages. As it allows for re-collection of new historical data every few months, and uses the new data to retrain and fine-tune the ACU control policy accordingly. This endows our system with high adaptability, providing an evolvable control optimization solution to a slowly changing industrial system.

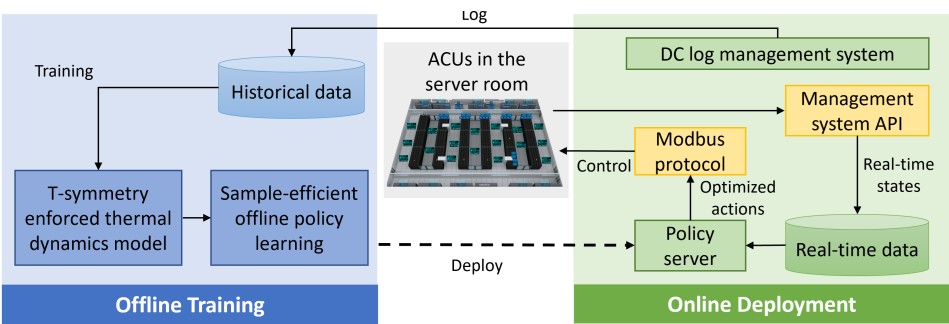

Figure 8: Overall architecture of the deployment system.

## B    REAL-WORLD TESTING ENVIRONMENTS AND EXPERIMENT SETUPS

### B.1    PRODUCTION DATA CENTER ENVIRONMENT

Figure 9 presents some photographs and the layout illustration of our real-world data center testing environment. In this large-scale commercial data center, we are granted permission to conduct experiments in two designated server rooms. These server rooms host the real IT loads of a large video-sharing website in China. Specifically, in Server Room A, the average total server load is around 550 kW, with an overall ACU power consumption of around 25 kW; in Server Room B, the average total server load is around 610 kW, and the overall ACU power consumption is about 37 kW. In the early stages of the experiment, we are only allowed to control 4 ACUs in each server room (ACU 1-6, 1-5, 2-5, and 2-4 on the left side in Server Room A; ACU 1-1, 1-2, 2-1, and 2-2 on the right side in Server Room B). These ACUs are arranged in pairs on opposite sides of the room, directly influencing two cold aisles and one hot aisle. The remaining ACUs continue to operate under PID control. After verifying the effectiveness of the experiments, we further used the model to control

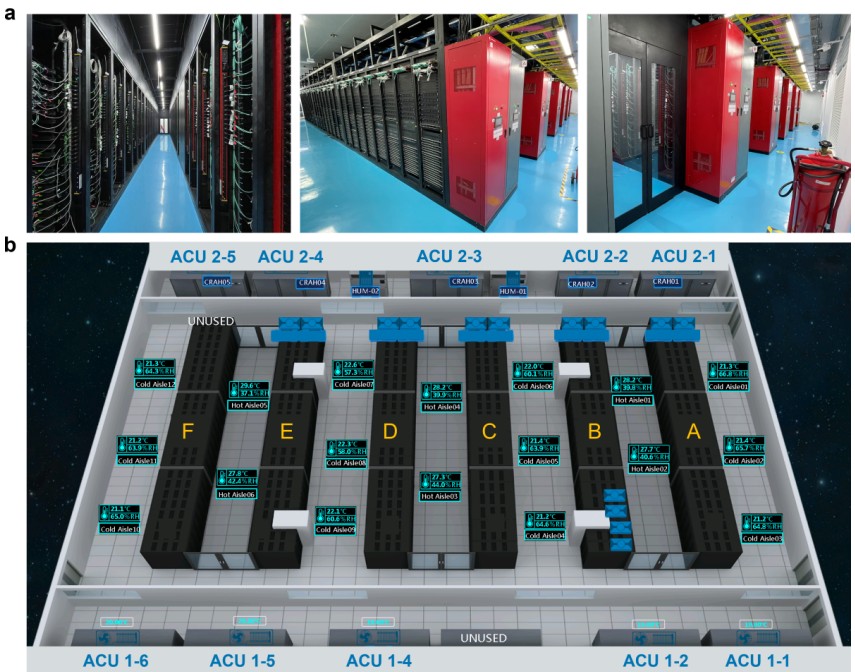

Figure 9: The photographs and layout illustration of the real-world commercial data center. **a**, Photographs of the interior of a server room, showcasing the hot aisle, cold aisle, and server racks from left to right. **b**, Overhead panoramic view of a server room, illustrating the spatial arrangement of all pertinent equipment.

6 ACUs in each server room (ACU 1-2, ACU 1-4, ACU 1-6, ACU 2-1, ACU 2-3 and ACU 2-4 in each server room). Finally, we conducted experiments controlling all ACUs in both server rooms (In Server Room B, all 10 ACUs are shown in Figure 9b. In Server Room A, there are 11 ACUs, with an additional ACU (ACU 1-3) located in the position marked as 'UNUSED' at the bottom of Figure 9b). As we are testing on the safety-critical real production environment, it is not possible for us to fully evaluate and test other baseline methods, as they may not have strong safety assurance. We leave these comparative experiments to our testbed environment, where we have full control.

We follow the DC industry's standard practice that specifies the target threshold of cold aisle temperature (CAT) as 25°C. For Server Room A, we collected about 20 months' historical operational data from the logging system, including approximately 180,000 data samples at 5-minute intervals, involving 108 state and action features. Similarly, for Server Room B, we collected historical data over 15 months, amounting to approximately 140,000 data samples, also at 5-minute intervals, and encompassing a total of 101 state and action features. The amount of real-world data available to train our offline RL policy is significantly fewer than typical offline RL benchmark tasks like D4RL (Fu et al., 2020) (often using 1 million data samples to learn simple tasks), especially considering the much larger scale of our problem. We run a series of offline policy evaluation tests and open-loop inspections to select the best-performing models and deploy them in real systems for closed-loop control evaluation. During this phase, our control system takes the real-time data from the cooling system every five minutes as inputs, then computes the optimized actions and directly transmits these commands to the ACUs for modifying fan speeds and valve opening percentage. We conducted a series of short and long-term experiments from January to December 2024. Our system has been operated safely for over 2000 hours. Through these comprehensive experiments, we verified that our proposed physics-informed offline RL framework and the resulting control system can operate both effectively and safely under the stringent safety and operational constraints of a real-world commercial data center.

## B.2 REAL-WORLD TESTBED

To thoroughly assess the performance of our proposed method, we also constructed a real-world testbed environment, which contains 22 servers and an inter-column air conditioner as the ACU

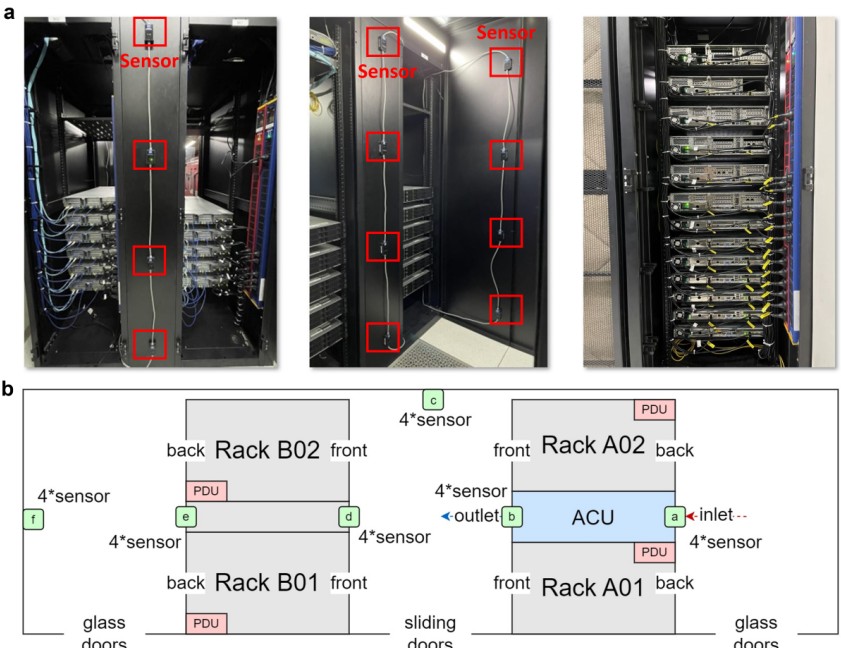

Figure 10: The photographs and layout illustration of our constructed small-scale DC testbed. **a**, Illustration of the installed temperature and humidity sensors in our testbed. **b**, Layout illustration of the testbed.

(located between Rack 1 and Rack 2). This is a compressor-based ACU, which is smaller than the typical ACUs in commercial data centers that use the cold water from chillers and cooling towers as the cold source. Therefore the fans and the compressor inside the ACUs are the primary contributors to the ACU's energy consumption. For the testbed environment, temperature regulation is achieved by adjusting the entering air temperature (EAT) setpoint of the ACU to ensure the CAT remains below the predetermined threshold. We installed 6 sets of temperature and humidity sensors (24 in total) to monitor the internal temperature field inside our DC testbed environment. Moreover, we also have access to the interior temperature sensor readings from each server, which provides even finer-grained monitoring of the thermal dynamics inside the testbed. Figure 10 provides a detailed depiction of the testbed environment configuration.

To support testing with a wide variety of server loads, we also developed a software framework to assign servers with different load patterns that mimic real-world IT tasks. The software employs a Kubernetes (k8s) cluster architecture and is implemented under the CentOS Stream 9 operating system. The ACU control is implemented through the Modbus protocol, which regulates the setpoint of the Entering Air Temperature (EAT) of the ACU, thereby indirectly adjusting the fan and compressor of the ACU. In our experiments, the control policies calculate and output the EAT setpoint every two minutes and control the ACU accordingly. The experimental server loads in our testbed range from approximately 5 to 8 kW, while the power consumption of ACU varies from 1.5 to 4 kW. We also built a data collection and database management system using InfluxDB and Telegraf to handle and store the real-time and historical data in our testbed. We collected the historical operational data over 61 days, comprising approximately 43,000 data samples at 2-minute intervals, comprising 105 state and action features.

As we have complete control over our testbed, we can conduct extensive exploratory experiments with our proposed method and compare it with a wide range of existing baseline methods without restriction. As temperature regulation in the DC testbed is comparably easier than in the large-scale production DC environment, we employed a stricter 22°C CAT threshold to make the control tasks more challenging. Furthermore, we also conducted experiments on the impact of weight coefficients on our reward function and carried out ablation studies to further evaluate our method.

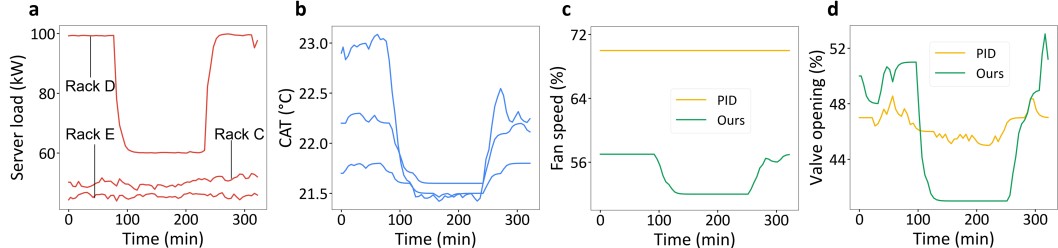

Figure 11: ACU control behaviors of our method and the PID controller under drastic server load fluctuation. **a**, Load variation pattern of three server racks (Rack C, D, E) during the selected time period, with one server rack having a drastic load drop and increase. **b**, Temperature readings from the three most relevant cold aisle sensors. **c**, **d**, The variations in fan speed and valve opening for two ACUs during the time period, with one controlled by the PID controller (ACU 1-1) and the other by our method (ACU 1-2).

## C  ADDITIONAL RESULTS

### C.1  PERFORMANCE UNDER DRASTIC SERVER LOAD FLUCTUATION

To further evaluate the adaptability and load-awareness of our method, we tested on a specific scenario with drastic server load fluctuations in Server Room B. We compare the control strategy of two ACUs with one controlled by the default PID controller and the other by our method. Experimental results are presented in Figure 11. The PID controller demonstrates limited adaptability in this scenario, with no adjustments to fan speeds and only marginal changes in valve opening percentage. In contrast, our offline RL approach was able to promptly adapt to external changes, resulting in a more optimal and energy-efficient strategy. These results underscore the effectiveness and adaptability of our approach in highly dynamic DC service conditions.

### C.2  ADDITIONAL ABLATIONS ON REWARD FUNCTION DESIGN

We considered both the control parameters of the ACUs and environmental factors within the cooling system to design a reasonable reward function for RL policy learning. For the weight coefficient $\beta_1, \beta_2, \beta_3, \beta_4$ in the reward function Eq. (1), we set their values as the reciprocal of the mean of the corresponding reward term calculated based on the preprocessed dataset. This ensures each reward term has a similar scale. For the first constant term $r_0$ in the reward function, to keep the reward positive, we calculate the sum of the other terms in the reward function for each record in the preprocessed dataset and take their maximum value plus 1 as the value of $r_0$.

To further investigate the robustness of our reward function design, we also conducted additional experiments on the testbed by varying the relative scale of the third term ($\beta_2 \sum_{n=1}^{N} \ln(1 + \exp(T_c^m - \rho_T))$) in Eq. (1), which controls the strength of CAT violation penalty. Specifically, we test the default value of $\beta_2$ as well as multiply it by 5 and 10, to test the impact of prioritizing more on safety constraint satisfaction. We train three models with different $\beta_2$ values and use the resulting models to control the ACU for 6 hours under low and high server load conditions on the testbed. In all these experiments, the CAT was controlled below the predefined threshold. Moreover, as reported in Table 2, the energy-saving performances of the models under different $\beta_2$ weight coefficients consistently achieve comparable and low ACLF values. This shows our designed reward function is robust and does need much tuning to ensure good practical performance, which is particularly desirable for real-world deployments.

Table 2: Performance on the testbed using different scale of $\beta_2$ in the reward function.

|  | Default $\beta_2$ | $5 \times \beta_2$ | $10 \times \beta_2$ |
| --- | --- | --- | --- |
| ACLF (%) under low server load | 29.66 | 29.37 | 30.95 |
| ACLF (%) under high server load | 26.89 | 27.50 | 26.05 |

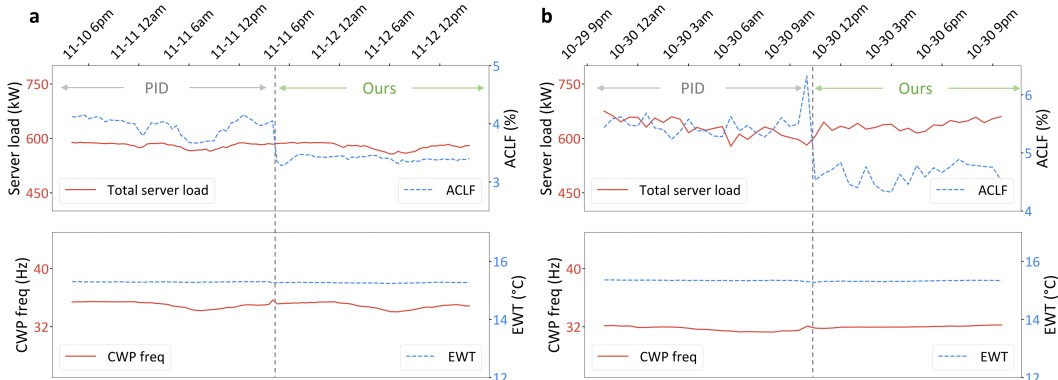

Figure 12: In the full control experiment of ACUs in the commercial data center, the water-side related states are as follows. **a**, In Server Room A, before and after the use of our offline RL policy for control, the overall server load remains stable, and both the chilled water pump frequency (CWP freq) and the ACUs' entering water temperature (EWT) also stay stable. After implementing our control policy, the ACLF value significantly decreases. **b**, In Server Room B, a similar comparison of the server load and water-side indicators before and after the use of our offline RL policy for control, which shows consistent results with those in part **a**.

## C.3 ANALYSIS OF IMPACTS ON THE UPSTREAM WATER-SIDE COOLING SYSTEM

To evaluate the potential impact of optimizing the air-side cooling system using our method on the upstream water-side cooling system, we conducted additional analysis on the water-side related states through two before-and-after tests. We select two time periods with relatively stable server loads in the two server rooms (November 10-11 for Server Room A and October 29-30 for Server Room B) to compare the chilled water pump frequency (CWP freq) and the ACUs' entering water temperature (EWT) before and after using our offline RL policy for control. The CWP freq. and EWT are key states that reflect the working conditions of the water-side cooling system, which are external factors to the air-side cooling systems. Figure 12 shows the experimental results during the full control of all ACUs in Server Room A and Server Room B. The Figure 12a and 12b, the dashed vertical lines indicate the time points when our method took over the control. In both Server Room A and Server Room B, before and after our method began controlling, the average EWT of the ACUs remained stable. Additionally, the chilled water pump frequency of the water-side cooling system also did not exhibit significant variations. However, comparing the results before and after adopting our control method, the ACLF values in both rooms significantly decreased. These results demonstrate that although our method effectively reduces the air-side cooling system's energy consumption, it does not have a noticeable impact on the upstream water-side cooling system. Moreover, as also shown in Section 4.1, Figure 3 and 4, our offline RL policy enables much better temperature regulation and forms a more stable temperature field for the hot aisles, due to smartly coordinating the control of all ACUs based on the dynamic temperature patterns in the server rooms. This effectively decreases the oscillation in the conventional control approach, which often results in frequent overshoots during temperature control and causes higher ACU energy consumption. This partly explains why our method can have lower energy consumption but achieve the same or better cooling effect.

## D  IMPLEMENTATION DETAILS

### D.1  PRACTICAL IMPLEMENTATIONS OF OUR PROPOSED METHOD

**Data preprocessing.** We preprocessed the DC raw data to facilitate model training. Min-max normalization was applied to both states and actions using the following formulas: $\tilde{s} = \frac{(s - S_{min})}{(S_{max} - S_{min})}$ and $\tilde{a} = \frac{(a - A_{min})}{(A_{max} - A_{min})}$. $S_{max}, S_{min}, A_{max}, A_{min}$ are maximum and minimum normalization boundaries for state and action features. For actions, we set $A_{min} = 0$ and $A_{max} = 100$ as both fan speed and valve openings are percentage values. For the states, as described in Section 3.1, there exist different types of sensor inputs: $s = \{s_s, s_a, s_e\}$, and each type of sensor reading has distinct scales.

Therefore, we set different normalization scales for different types of sensors by consulting the domain experts. Specifically, we denote all temperature-related sensor readings as $s_{temp}$ (e.g., LAT($s_a$), EAT($s_a$), LWT($s_a$), and EWT($s_e$)), humidity sensor readings as $s_{humi}$, and power consumption of servers as $s_{power}$. Their corresponding normalization boundaries are presented in Table 3.

Table 3: Normalization boundaries for different state components.

|           | $s_{temp}$ | $s_{humi}$ | $s_{power}$ |
|-----------|------------|------------|-------------|
| $S_{min}$ | 1          | 0          | 0           |
| $S_{max}$ | 40         | 100        | 150         |

**Model architecture and hyperparameters.** The architecture and algorithm hyperparameters in our proposed physics-informed offline RL framework are listed in Table 4. As discussed in Section 3.3, the only hyperparameter that we tuned during our experiments is $\alpha$ in the normalization term $\lambda_\alpha$ (see Eq. (9)). We tuned $\alpha$ values in the range of $[2.5, 10]$ and deployed the best-performing model for long-term control in both the production DC and our testbed environments. This hyperparameter modulates the conservatism of the learned policy. We observe that reasonably increasing $\alpha$ can enhance the energy-saving performance to a certain degree.

**Real-time data preprocessing and policy smoothing.** In our deployed systems, we preprocess the real-time sensor data by filtering out problematic data samples and resample them into uniform time intervals (5 minutes for the large-scale commercial data center and 2 minutes for the small-scale DC testbed). To enhance the smoothness and robustness of the closed-loop ACU control commands generated by the policy, we apply temporal smoothing to the policy-generated actions in our practical implementation. Specifically, the final execution action at the current time step is calculated as the average of policy output actions at the current time step and the previous 4 time steps, which provides a smoother control signal for ACUs.

**Algorithm pseudocode.** The pseudocode of our proposed physics-informed offline RL framework can be found in Algorithm 1.

---

**Algorithm 1**

---

**Require:** Preprocessed historical dataset $\mathcal{D}$, initialized value network $Q$, policy network $\pi$, and the T-symmetry enforced thermal dynamics model (TTDM), which contains the state-action encoder $\phi(s, a)$, latent forward dynamics model $f$ and latent reverse dynamics model $g$, state and action decoders $\psi(z_s)$ and $\psi(z_a)$.
    *// Learning TTDM from offline dataset*
    **for** $t = 1, \cdots, T_1$ training steps **do**
        Sample a mini-batch $B$ of samples $\{(s, a, s', a')\} \sim \mathcal{D}$ and process through the state-action encoder $\phi(s, a)$ to get the latent representations $\{(z_s, z_a, z_{s'}, z_{a'})\}$.
        Compute the forward and reverse dynamic losses based on Eq. (3) and Eq. (4)
        Compute T-symmetry regularization loss over the two latent dynamics models based on Eq. (6)
        Compute the reconstruction losses in Eq. (2) and Eq. (5)
        Update TTDM network parameters by minimizing the overall learning objective in Eq. (7)
    **end for**
    *// Sample efficient offline policy optimization*
    **for** $t = 1, \cdots, T_2$ training steps **do**
        Sample a mini-batch $B$ of samples $\{(s, a, r, s')\} \sim \mathcal{D}$, where $r$ is calculated based on Eq. (1)
        Update the value network $Q$ with the learned $\phi(s, a)$ based on the objective in Eq. (8).
        Update the policy $\pi$ based on the policy learning objective in Eq. (9).
    **end for**

---

## D.2 BASELINE ALGORITHMS

In our testbed experiments, we compare our method with the ACU's default PID controller, a data-driven MPC method for DC cooling control developed by Google (Lazic et al., 2018), an off-policy RL-based DC cooling optimization method CCA (Li et al., 2019), mainstream offline RL methods

Table 4: Hyperparameter details.

|  | Hyperparameters | Value |
|---|---|---|
| TTDM Architecture | Optimizer type | Adam |
|  | Learning rate | 3e-4 |
|  | Weight decay | 1e-5 |
|  | Channel number | 6 |
|  | Common feature per node | 4 |
|  | GNN hidden layers | 2 |
|  | GNN hidden units | 256 |
|  | Forward / reverse model hidden layers | 2 |
|  | Forward / reverse model hidden units | 128 |
|  | Fusion layers | 2 |
|  | Fusion layer units | 128 |
|  | Weight of $\ell_{T-sym}$ and $\ell_{rec}$ | 1 |
|  | Weight of $\ell_{rvs}$ and $\ell_{fwd}$ | 0.1 |
| Offline RL Algorithm | $\alpha$ | Tuned in the range of [2.5,10] |
|  | Discount factor $\gamma$ | 0.99 |
|  | Target update rate | 0.005 |
|  | Policy noise | 0.2 |
|  | Critic neural network layer width | 512 |
|  | Actor neural network layer width | 512 |
|  | Actor learning rate | 3e-4 |
|  | Optimizer type | Adam |
|  | Critic learning rate | 3e-4 |
|  | Policy noise clipping | 0.5 |
|  | Policy update frequency | 2 |
|  | Number of iterations | 5e5 |

such as Implicit Q-Learning (IQL) (Kostrikov et al., 2022) and Conservative Q-Learning (CQL) (Kumar et al., 2020), and the state-of-the-art (SOTA) safe offline RL algorithm, FISOR (Zheng et al., 2024). We provide detailed descriptions of these baseline methods as follows.

**Default PID controller.** The ACU in our experiments adopts a conventional PID controller (Ang et al., 2005) to adjust its fan speed and compressor to minimize the error between the target CAT setpoint and the system's actual CAT value. The controller consists of three components: the **P**roportional term, which responds to the current error; the **I**ntegral term, which accumulates past errors to correct steady-state offsets; and the **D**erivative term, which predicts future errors based on the rate of change.

**Data-driven MPC controller** (Lazic et al., 2018). This DC cooling control method is developed by Google, which learns a linear dynamics model of the floor-level cooling system for future state prediction, and optimizes the control action over a finite time horizon using MPC. At each time step, MPC solves a constrained optimization problem to minimize a cost function while considering system constraints.

**Cooling Control Algorithm (CCA)** (Li et al., 2019). CCA is an actor-critic RL framework for DC cooling control. It is based on the classic off-policy RL algorithm deep deterministic policy gradient (DDPG) (Lillicrap et al., 2015). As DDPG is an online RL method, CCA needs online interactions with a simulation environment to collect and store data in a replay buffer, and sample training batches from the replay buffer for policy learning. In our offline learning setting, as there is no reliable simulation environment available, we replace CCA's replay buffer to the offline dataset in our implementation.

**Implicit Q Learning (IQL)** (Kostrikov et al., 2022). IQL is a popular offline RL algorithm that uses expectile regression to learn value functions from fixed datasets without explicit policy constraints. It avoids evaluating the potential OOD actions from the learned policy, therefore alleviating distributional shift, and typically enjoys stable offline policy learning.

**Conservative Q learning (CQL)** (Kumar et al., 2020). CQL is another popular offline RL algorithm that learns conservative estimates of Q-values on OOD actions to enforce offline behavioral data regularization and mitigate distribution shifts.

**Feasibility-guided Safe Offline RL (FISOR)** (Zheng et al., 2024). FISOR is the SOTA safety-centric offline RL algorithm which enforces hard constraints by identifying the largest feasible region from the offline dataset based on Hamilton-Jacobi (HJ) reachability analysis. It adopts a decoupled learning scheme that optimizes a diffusion model-based safe policy by maximizing reward within the feasible regions while minimizing safety violations within infeasible regions, thereby enjoying a strong safety performance and superior learning stability.

## E    REAL-WORLD DATA ANALYSIS

In the real-world data center, due to the use of PID group control for ACUs throughout the historical operation, and infrequent adjustments to the PID-related temperature setpoints, the action patterns of the ACUs system (fan speed and water valve opening) are narrowly distributed. Additionally, the distributions of other state features are mostly concentrated with a single peak. All these factors pose significant challenges to offline RL policy learning, requiring models with strong generalization capability to effectively learn and optimize control strategies. Figure 13 shows the historical dataset distributions collected from our real-world testbed, in which we collect system operational data from more diverse server load and control settings, resulting in relatively broader state-action space coverage. This actually makes the task more manageable for existing offline RL algorithms like CQL, IQL, and FISOR. However, as we have shown in Figure 6, our proposed method still outperforms the baseline methods in the testbed experiments, and more importantly, achieves good performance in the much more challenging production DC environment.

## F    LIMITATIONS AND FUTURE WORKS

In this study, we only tested in a single large-scale commercial DC facility and a small-scale real-world testbed. For future works, we plan to further expand our experiments to multiple DC facilities with different air-side cooling system configurations. Also, our approach models the safety constraints by incorporating them as penalty terms inside the RL reward function, which adds complexity to reward design and may not be sufficient to ensure safety under certain special conditions. Future investigations can be conducted to expand our method to a safe offline RL framework, with dedicated consideration of constraint satisfaction, which would provide more safety guarantees in practice. Furthermore, it is also meaningful to explore the joint optimization of both cooling and server-side systems, which can fully maximize the potential for DC energy saving.

## G    LEARNING CURVES

Figure 14 reports the learning curves of the proposed TTDM and offline policy learning method. As it is not possible to directly interact with the real DC environment and evaluate the policy's performance during offline RL training, hence we report the Q-function learning loss and policy loss for different training steps. Both our proposed TTDM and the RL policy learning scheme enjoy stable model convergence during training.

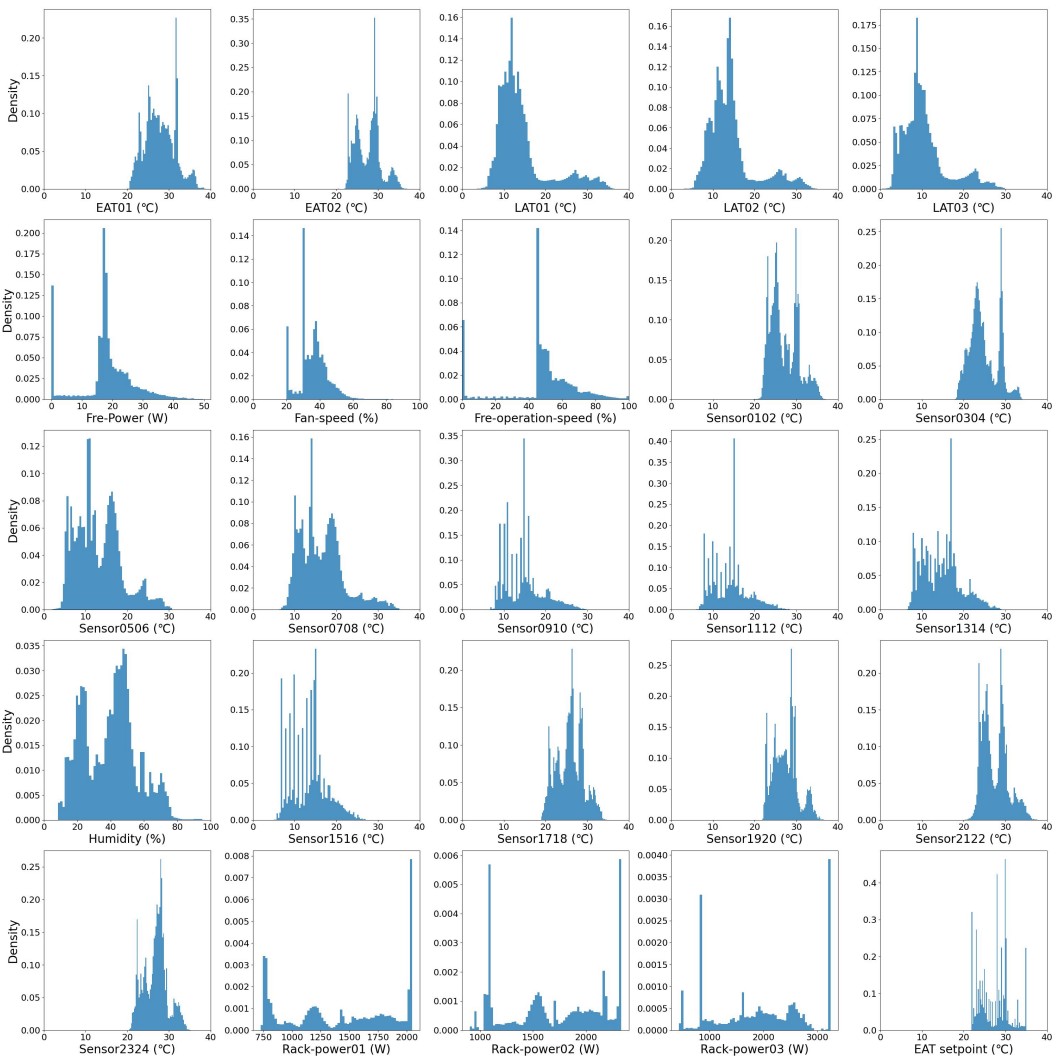

Figure 13: Distributions of the state and action features in our historical dataset collected from the real-world DC testbed.

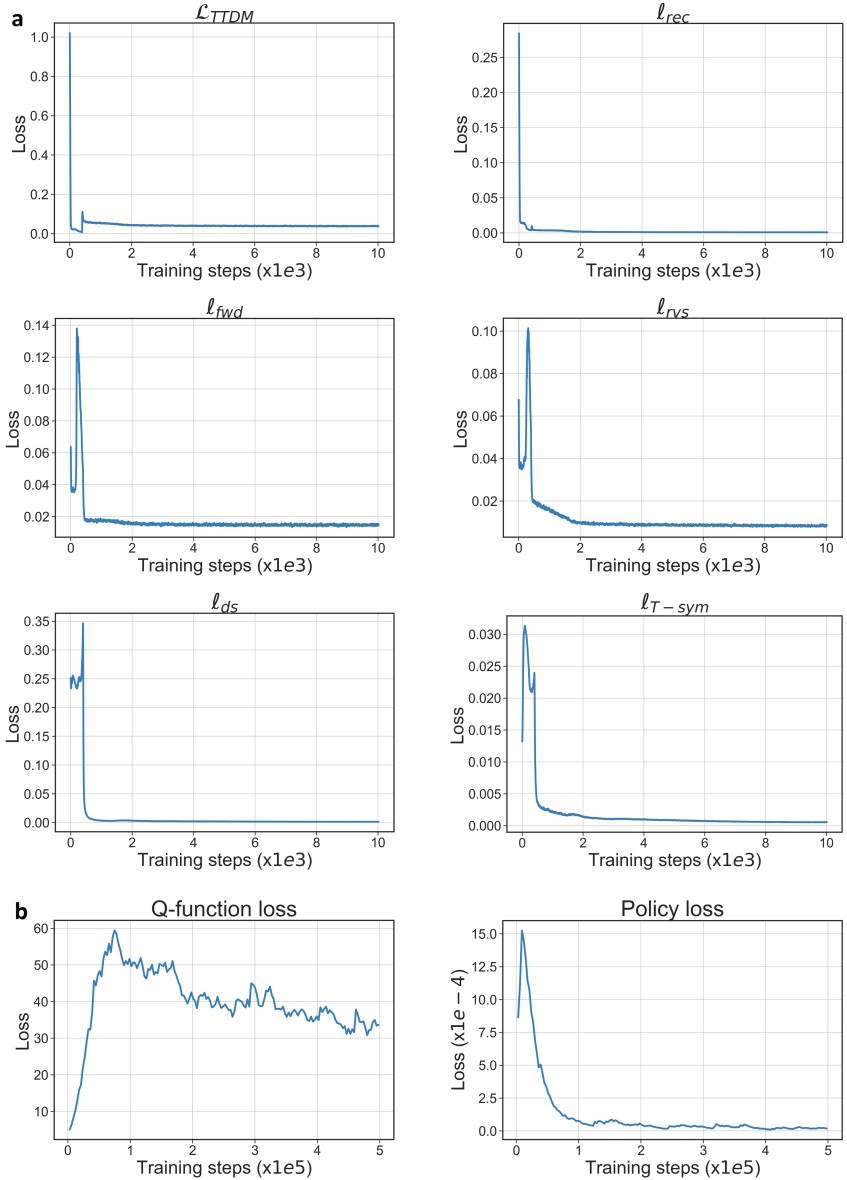

Figure 14: **a**, Learning curves of the overall loss function and each individual loss term of TTDM. **b**, Learning curves for the offline RL policy learning.

