# OpenReview forum: "Data Center Cooling System Optimization Using Offline Reinforcement Learning"
_ICLR.cc/2025/Conference — ICLR 2025 Poster_

### Official Review · Reviewer_FhGh · 2024-10-31

**Soundness:** 3
**Presentation:** 3
**Contribution:** 3
**Rating:** 6
**Confidence:** 4

**Summary:**

The paper introduces a physics-informed offline reinforcement learning framework for optimizing data center (DC) cooling systems, addressing their substantial energy demands. Utilizing a graph neural network (GNN) architecture with time-reversal symmetry (T-symmetry), the model effectively captures complex thermal dynamics and ensures robust policy learning from limited historical data. Deployed in a real-world DC, the framework demonstrated 14-18% energy savings over 1300 hours without safety violations. Validation in a testbed environment further confirmed its superior performance over conventional and baseline RL methods. This approach highlights offline RL's potential for data-limited, safety-critical industrial control applications beyond DCs.

**Strengths:**

- The paper integrates graph neural networks (GNNs) with time-reversal symmetry (T-symmetry) to enhance offline reinforcement learning (RL), showcasing an innovative approach to optimizing DC cooling systems.
- Comprehensive real-world validation includes 1300 hours of deployment in a production DC, proving the method’s effectiveness and robustness.
- The methodology is clearly detailed with supporting figures and tables, making the framework’s architecture and results easy to follow.
- Demonstrated 14-18% energy savings highlights significant practical impact, with potential for broader application to other data-limited, safety-critical industrial control scenarios.

**Weaknesses:**

- The paper relies solely on simulation-based testing in the small-scale DC testbed environment, limiting the generalizability of the results to more diverse real-world DCs. While the full-scale production test demonstrates feasibility, more varied test environments could strengthen the claims.
- The deployment details and considerations for long-term adaptability, such as how frequently model retraining is needed or how it adapts to evolving DC conditions, are underexplored.
- The scalability of the approach for larger DCs or environments with higher variability is not deeply discussed. Addressing the implications for larger-scale or more complex DC configurations would improve the scope of the work.
- While T-symmetry and GNN integration are well-motivated, there is limited discussion on potential trade-offs, such as computational complexity or latency during model training and execution.
- The paper could benefit from including a comparison with relevant existing methods like CLUE[1], which demonstrated data-efficient HVAC control with only seven days of training data and reduced comfort violations. Such a comparison would provide valuable context regarding the data efficiency and performance trade-offs of the proposed framework, particularly under conditions of limited training data.
1. An, Zhiyu, Xianzhong Ding, Arya Rathee, and Wan Du. "Clue: Safe model-based rl hvac control using epistemic uncertainty estimation." In Proceedings of the 10th ACM International Conference on Systems for Energy-Efficient Buildings, Cities, and Transportation, pp. 149-158. 2023.

**Questions:**

- Could the authors elaborate on how their model adapts to changing DC conditions over time and how often retraining is needed to maintain optimal performance?
- What are the potential limitations or challenges when scaling the proposed approach to larger data centers or DCs with more complex cooling system architectures?
- Can the authors provide more details on the computational cost of deploying the GNN and T-symmetry framework, particularly in comparison to simpler baseline models like PID controllers?
- Was the impact of T-symmetry enforcement on training time and model convergence evaluated? It would be helpful to understand if this enhancement has significant trade-offs in terms of training efficiency.
- Given that CLUE [1] achieved effective HVAC control with only seven days of training data, have the authors considered comparing their method's data efficiency to similar approaches? How would their framework perform with similarly limited training data?
1. An, Zhiyu, Xianzhong Ding, Arya Rathee, and Wan Du. "Clue: Safe model-based rl hvac control using epistemic uncertainty estimation." In Proceedings of the 10th ACM International Conference on Systems for Energy-Efficient Buildings, Cities, and Transportation, pp. 149-158. 2023.

---

> ### Author Response · Authors · 2024-11-20
> **Response to Reviewer FhGh (Part 1/2)**
>
> We thank the reviewer for the comments. Regarding the concerns of the reviewer, we provide the following responses.
>
> > **W1. The paper relises solely on simulation-based testing in the small-scale DC testbed environment...**
>
> As we have clearly stated in the abstract, introduction, and experiment sections, we built a **real-world DC testbed environment** rather than a simulation environment, it has 22 real servers and an ACU. We also installed a series of temperature and humidity sensors to monitor the environmental states. All our experiments are conducted in the real-world environment, with completely no simulation involved. We refer the reviewer to Appendix B.2 of our paper, which provides a detailed description of our real-world testbed environment.
>
> > **W2. The deployment details and considerations for long-term adaptability, such as how frequently model retraining is needed or how it adapts to evolving DC conditions, are underexplored.**
>
> We actually discussed it in Appendix A, Lines 723-728 of our paper. We recommend the practitioners to re-collect new historical data every few months (e.g., 3-6 months), and use the new data to retrain and fine-tune the ACU control policy accordingly. This endows our system with high adaptability, providing an evolvable control optimization solution to a slowly changing industrial system.
>
> > **W3. The scalability of the approach for larger DCs or environments with higher variability is not deeply discussed. Addressing the implications for larger-scale or more complex DC configurations would improve the scope of the work.**
>
> - The commercial DC selected to conduct our real-world experiments is actually one of the flagship DCs in China, which is larger than most commercial DCs. Its server rooms contain 120 server racks (more than 2000 servers in a single room) and 10-11 ACUs. We are not sure what the reviewer meant by testing for a large-scale and more complex DC configuration, as our experiments are already conducted on a flagship large-scale commercial DC that has complex air-side cooling systems.
> - In Figure 11 and Appendix C.1, we actually conducted experiments to evaluate our method under drastic server load fluctuations. We compare the control strategy of two ACUs with one controlled by the default PID controller and the other by our method. The results show that our offline RL method was able to promptly adapt to external changes as compared to the default PID controller, resulting in more optimal and energy-efficient control.
>
>
> > **W4. While T-symmetry and GNN integration are well-motivated, there is limited discussion on potential trade-offs, such as computational complexity or latency during model training and execution.**
>
> - In our approach, both the TTDM and RL training phases involve T-symmetry and GNN integration. However, the policy network obtained after training does not involve these two components. The final model deployed in the experimental environment is only the policy network, which is a simple MLP network and can be efficiently evaluated during inference. Moreover, our proposed TTDM model only contains lightweight GCN and MLP layers, which does not incur a significantly increased computational cost as compared to many other existing methods that use heavy architectures like Transformer or diffusion models.
> - The table below lists the training time and the inference time for the different components of our model used in the commercial DC experiments. The model training is conducted on a workstation with AMD Ryzen 7950X 16-core CPU, 64G memory, and Nvidia RTX4090 GPU. The policy network is very light and its inference can run on most CPU machines.
>
> | Component | TTDM training | Offline RL training | Policy inference (1 step) |
> | -------- | -------- | -------- | -------- |
> | Duration | 5 hours  | 4 hours  | 0.1 ms |

---

> ### Author Response · Authors · 2024-11-20
> **Response to Reviewer FhGh (Part 2/2)**
>
> > **W5.The paper could benefit from including a comparison with relevant existing methods like CLUE [1].**
>
> We thank the reviewer for providing this reference, and we have already included it in the related work of our revised paper. However, we also would like to point out that the method proposed in CLUE [1] is not directly applicable to our air-side cooling system optimization problem, specifically,
> - CLUE is proposed to optimize the temperature control of a zone with an HVAC. However, our air-side DC cooling system optimization problem involves 10-12 ACUs and complex thermal dynamics patterns inside the server room, due to frequently changing sever loads and physical locations of servers. This problem actually needs joint control of multiple ACUs in a way that is fully load-aware and capable of capturing complex thermal dynamics. The problem setup is completely different from the one studied in CLUE, and is much more complex.
> - The action in CLUE is the HVAC heating and cooling setpoints, however, this is not applicable to our DC optimization problem. This is because all ACUs in the data center use the same cold aisle temperature setpoint (25°C), and this value never changes in the historical operational data. The PID controller of ACUs adjusts the low-level control (fan speeds and valve openings) to follow this temperature setpoint. Hence if the action to be optimized is the temperature setpoint, then it is basically impossible to learn any meaningful RL policy, since all actions have the same value in the offline dataset. Instead, in this work, we optimize the fan speeds and valve openings inside the ACU, which directly achieves low-level control of ACUs.
> - Lastly, CLUE is only validated in the EnergyPlus simulation environment rather than in the real-world environment. By contrast, all our evaluations are conducted in real-world systems.
>
>
> > **Q1. Could the authors elaborate on how their model adapts to changing DC conditions over time and how often retraining is needed to maintain optimal performance?**
>
> Please refer to our response to W2.
>
> > **Q2. What are the potential limitations or challenges when scaling the proposed approach to larger data centers or DCs with more complex cooling system architectures?**
>
> Please refer to our response to W3.
>
>
> > **Q3. Can the authors provide more details on the computational cost of deploying the GNN and T-symmetry framework, particularly in comparison to simpler baseline models like PID controllers?**
>
> > **Q4. Was the impact of T-symmetry enforcement on training time and model convergence evaluated?**
>
> - Please refer to our response to W4. The policy inference of our method only takes about 0.1ms, which does not have any computational issues during deployment.
> - In Appendix G of our revised paper, we reported all the learning curves of different components of our method. Both our proposed TTDM and the RL policy learning enjoy good model convergence. Please check Appendix G of our paper for detailed plots.
>
> > **Q5. Given that CLUE [1] achieved effective HVAC control with only seven days of training data, have the authors considered comparing their method's data efficiency to similar approaches? How would their framework perform with similarly limited training data?**
>
> Please refer to our response to W5.

---

> > ### Comment · Reviewer_FhGh · 2024-11-26
> >
> > Thank you for your response! I’ve decided to increase my score by one point

---

> > > ### Author Response · Authors · 2024-11-26
> > > **Thank you for raising the score**
> > >
> > > Dear Reviewer FhGh,
> > >
> > > Thank you for raising the score! Please let us know if you have any remaining comments or feedback. We will be happy to address them.
> > >
> > > Authors of Submission 1285.

---

> ### Author Response · Authors · 2024-11-24
> **A gentle reminder for reviewer**
>
> Dear reviewer FhGh,
>
> As the rebuttal phase is coming to a close, we wanted to check back to see whether you have any feedback or remaining questions. **We would be happy to clarify further, and grateful for any other feedback you may provide.**
>
> We really appreciate your time engaged in the review process and look forward to your replies!
>
> Best regards,
>
> Authors of Paper 1285

---

### Official Review · Reviewer_4H4V · 2024-11-03

**Soundness:** 3
**Presentation:** 4
**Contribution:** 3
**Rating:** 8
**Confidence:** 5

**Summary:**

This paper presents a novel physics-informed offline reinforcement learning framework for optimizing energy efficiency in data center cooling systems. The core component is a T-symmetry enforced thermal dynamics model using graph neural networks to capture complex thermal patterns. This enables sample-efficient offline policy learning from limited historical data. The framework was successfully deployed in a large-scale commercial data center, controlling multiple air cooling units and achieving 14-18% energy savings without safety violations over 1300 hours of experiments. Comprehensive evaluations on a real-world testbed further demonstrated the method's effectiveness compared to baselines. The approach shows significant potential for solving data-limited, safety-critical industrial control problems beyond just data center cooling.

**Strengths:**

- The methods proposed have been deployed in a real-world data center. This is a big, big accomplishment
- The proposed methods go beyond simply applying an existing algorithm. They have applied state-of-the-art offline RL methods to the data center cooling problem. The idea to use both a GNN and a physics informed loss function makes sense.
- The T-symmetry is an interesting idea that has been adopted from another paper, and they show it works well in their setting.
- They have performed controlled experiments in a large scale data center as well as a small one where they have more control over the settings.
- Their evaluation includes comparison against the state-of-the-art as well as ablation of their algorithm

**Weaknesses:**

- The key metric used in evaluation is ACLF -- Air-side cooling load factor. However, this metric is neither defined nor explained in detail. It appears to increase with server load based on the sentence -- "Due to the smaller scale of the testbed and significantly lower server
load as compared to the real-world DC, the calculated ACLF values are higher than those observed in the real DC experiments."
- The Air Cooling Units (ACU) are being optimized in isolation, and it is unclear what is the impact on the upstream cooling system demand. Specifically, it is clear that the algorithm reduces the CAT (cold aisle temperature?) and therefore increases the demand on the chiller and cooling towers. Is the algorithm simply increasing the energy demand in the upstream system while reducing it in the ACUs?
- There are many claims throughout the paper which are not substantiated. For example: it states: "building a high-fidelity simulator can be very costly and impractical." How expensive is it, and why is it impractical? Another example: "the fan power consumption is proportional to the cube of the fan speed". Please cite the source of this information.
- There are two externalities to the system -- entering water temperature and server load. While server load has been accounted for in the experiments, it is unclear if entering water temperature is constant throughout the experiment.

**Questions:**

1. Why does the reward need to be positive?
2. How did you tune the hyper-parameters?
3. T-symmetry is supposed to help with OOD generalization. Have you measured OOD in your dataset?
4. The algorithm is based on TD3+BC. Why is TD3+BC not one of the baselines in your experiments?
5. I'm surprised outside weather conditions is not considered in the modeling. Does the data center have perfect insulation? I would assume hotter conditions would lead to higher cooling demand.
6. The ratio of ACU average electric power and average energy consumption seem to vary with each entry. Why would that be?

---

> ### Author Response · Authors · 2024-11-20
> **Response to Reviewer 4H4V (Part 1/2)**
>
> We sincerely appreciate the reviewer for the positive feedback and constructive comments. Regarding the concerns of the reviewer, we provide the following responses.
>
> > **W1: The explanation of ACLF is not defined nor explained.**
>
> - We apologize for the potential confusion regarding the definition of the Air-side Cooling Load Factor (ACLF). We actually defined the ACLF in the text of Section 4.1 (see Lines 369-372), which is calculated as the ratio of the ACU system’s energy consumption to the servers’ energy consumption during the test period. A lower ACLF value indicates higher energy efficiency for the cooling system. To make the definition more apparent, we also mentioned the definition of ACLF in the caption of Table 1 in our revised paper.
> - ACLF can be perceived as the ACU energy consumption normalized by servers' energy consumption. In real-world scenarios, as server load increases, the cooling demand for the ACU system also increases, which leads to higher energy consumption from the cooling system to avoid overheating risks. The reason that our testbed environment has higher ACLF values is because the number of servers as well as their energy consumption is much lower as compared to the production DC environment, and the ACU energy consumption takes a larger proportion of the the total power consumption in the server room, hence leads to higher ACLF values.
>
>
> > **W2. Impact on the upstream cooling system**
>
> - We thank the reviewer for this constructive comment. We have added Appendix C.3 and Figure 12 to discuss the impact of our control method on the upstream water-side cooling system. Specifically, we conducted additional analysis on the chilled water pump frequency (CWP freq.) and ACUs’ entering water temperature (EWT) through two before-and-after tests in different server rooms of the commercial data center. The CWP freq. and EWT are key states that reflect the working conditions of the water-side cooling system, which are external factors to the air-side cooling systems. As reported in Figure 12 of our revised paper, before and after our method began controlling, the average EWT of the ACUs and the chilled water pump frequency remained relatively stable, without exhibiting noticeable changed patterns. However, comparing the results before and after adopting our control method, the ACLF values in both rooms significantly decreased. These suggest that our method won't have a large impact on the upstream water-side cooling system.
> - Moreover, as also shown in Section 4.1, Figures 3 and 4, our offline RL policy enables much better temperature regulation for the hot aisles, due to smartly coordinating the control of all ACUs based on the dynamic temperature patterns in the server rooms. This effectively decreases the oscillation in the conventional control approach, which often results in frequent overshoots during temperature control and causes higher ACU energy consumption. This partly explains why our method can have lower energy consumption but achieve the same or better cooling effect.
>
>
> > **W3. Explaination on the claims on the cost of high-fidelity simulation and the relationship between power consumption and fan speed**
>
> We thank the reviewers for these comments.
> - Regarding the cost of building high-fidelity simulators for DCs. Currently, the most accurate approach for thermal dynamics simulation in a DC environment is through computational fluid dynamics (CFD) modeling. Typically, in order to build such a high-fidelity simulator for a server room, one needs to go through a nuanced modeling process to reproduce every geometric and ACU equipment detail, even small discrepancies in the material type or location of wind deflector are likely to cause noticeable difference in the simulated results. Typically, it requires an expert to carefully calibrate the configurations in expensive commercial simulation software for 2-4 weeks for a single server room, but still suffers from some unavoidable sim-to-real gaps. This is obviously unscalable given that even in a single commercial DC, there are lots of server rooms. By contrast, our proposed offline RL framework does not need to build any simulation environment, and can use the same algorithm to achieve energy saving for different server rooms.
> - As for "the fan power consumption is proportional to the cube of the fan speed", this is actually called the cube law or fan law in the HVAC systems [1, 2, 3], which is a widely used fact in the HVAC and DC industry.
>
>
> [1] https://www.servicefolder.com/resources/hvac-blog/hvac-fan-law.html
>
> [2] https://www.energystar.gov/products/data_center_equipment/5-simple-ways-avoid-energy-waste-your-data-center/replace-standard
>
> [3] https://hvacrschool.com/the-3-fan-laws-and-fan-curve-charts/

---

> ### Author Response · Authors · 2024-11-20
> **Response to Reviewer 4H4V (Part 2/2)**
>
> > **W4: Entering water temperature is constant throughout the experiment.**
>
> Yes, close to constant. In most large commercial data centers, the upstream water-side cooling system is controlled to provide cold water with a fixed target temperature. In commercial DC where we conduct our experiments, the temperature of the cold water that enters the ACUs is controlled at around 14.5°C, varying no more than 0.5°C.
>
> > **Q1. Why does the reward need to be positive**
>
> This is more of a design choice rather than a strict requirement. It is also fine if we formulate the rewards to be all negative, as long as they are not sometimes positive and sometimes negative. As we find in our empirical experiments positive and negative rewards often cancel out with each other, causing potential RL training instability.
>
> > **Q2. How did you tune the hyper-parameters?**
>
> We only tuned the the $\alpha$ hyperparameter in Eq.(9) with the choices of {2.5, 5, 7.5, 10} in our experiments. We train offline RL policies with each of these hyperparameter values, and go through an open-loop inspection  (inspecting the policy outputs with real-time data input, but without direct control) to identify the best policy for closed-loop deployment. We also discussed this in Appendix D.1 "Model architecture and hyperparameters" Section.
>
> > **Q3. T-symmetry is supposed to help with OOD generalization. Have you measured OOD in your dataset?**
>
> - In our DC cooling system optimization problems, there are about 100 states and 20-22 continuous actions, but we can only collect 140k-180k historical operational data (about 15~18 months). The data size is quite small compared to the scale of the control optimization problem. By comparison, typical offline RL benchmarks like D4RL often use 1 million data samples to learn simple tasks like MuJoCo (typically fewer than 30 states and actions combined). The complexity of our problem is far beyond the D4RL tasks but with much smaller datasets. Under such a small dataset setting, most of the state-action space actually becomes OOD.
> - In Figures 13 and 14 of Appendix F, we plot all the state and action feature distributions in our collected historical dataset of Server Room A in the real-world DC. Due to the use of PID group control strategy for ACUs throughout the historical operation, and fixed PID target temperature setpoints, the action patterns of the ACUs system (fan speed and water valve opening) are narrowly distributed. Additionally, the distributions of other state features are mostly concentrated with a single peak. All these factors pose significant challenges to offline RL policy learning, requiring the model to have strong generalization capability to achieve good performance.
>
> > **Q4. The algorithm is based on TD3+BC. Why is TD3+BC not one of the baselines in your experiments?**
>
> Our algorithm is actually closer to TSRL [4] rather than TD3+BC. The only similarity of our method and TSRL as compared to TD3+BC is the use of behavior cloning penalty in the policy optimization objective (Eq. (9)). However, our method and TSRL also introduce T-symmetry regularization in the policy optimization objective, and optimize in the latent space with T-symmetry enforcement design. These are already quite different from TD3+BC. As reported and compared in the TSRL paper [4], without the T-symmetry related representation and policy constraint, the TD3+BC performs quite badly under the small dataset setting, due to solely relying on the over-conservative BC policy constraints. We didn't include TD3+BC in our testbed experiment, as TD3+BC is a rather weak baseline, which is less performant than newer methods like IQL and FISOR.
>
> [4] Look beneath the surface: Exploiting fundamental symmetry for sample-efficient offline RL. NeurIPS 2023.
>
>
> > **Q5. I'm surprised outside weather conditions is not considered in the modeling.**
>
> The outside weather condition mostly impacts the upstream water-side cooling system rather than the downstream air-side cooling system (the focus of our paper). Most DC server rooms are isolated within the DC building, and the only external factor is the cold water provided by the upstream water-side cooling system, which is captured in the entering water temperature (EWT) in our model's states.
>
>
> > **Q6. The ratio of ACU average electric power and average energy consumption seem to vary with each entry. Why would that be?**
>
> Please refer to our response to W1.

---

> > ### Comment · Reviewer_4H4V · 2024-11-21
> > **Thank you for the detailed responses**
> >
> > I'll keep my current positive score.

---

> > > ### Author Response · Authors · 2024-11-22
> > > **Thanks for the positive feedback on our work!**
> > >
> > > Dear reviewer 4H4V,
> > >
> > > Thank you for the positive feedback and recognition of our work. We really appreciate the effort you put into the review phase. Thanks again!

---

### Official Review · Reviewer_JenY · 2024-11-04

**Soundness:** 3
**Presentation:** 3
**Contribution:** 2
**Rating:** 8
**Confidence:** 3

**Summary:**

This paper proposes physics informed offline RL framework to control datacenter cooling systems. The proposed framework constructs a dynamics model based on T-symmetry and Graph Neural Networks to embed domain knowledge, TD3-BC to perform the policy optimization.  The framework is deployed on a real system and the authors develop a test bed to compare with existing approaches.

**Strengths:**

1. The authors tackle a problem of significant relevance, and develop a framework that can be implemented in the real world.
2. The paper is generally well written, and the authors do a good job of testing the proposed framework under different conditions.

**Weaknesses:**

1. The algorithmic contribution is minimal. While effective, the proposed method is a combination of existing methods tailored to a specific use case.

**Questions:**

1. Apart from the reward function, what are the other measures taken to prevent safety violations?
2. Have the authors tried Offline-Online methods? wherein the policy trained using offline RL deployed is constantly improved using new data obtained?
3. It seems like CQL does perform better than the proposed method in Figure 6, but at the cost of safety. What was the reward function used for CQL?

---

> ### Author Response · Authors · 2024-11-20
> **Response to Reviewer JenY (Part 1/2)**
>
> We really appreciate the reviewer for the constructive comments and positive feedback on our paper. Regarding the concerns of the reviewer, we provide the following responses.
>
> > **W1. The proposed method is a combination of existing methods tailored to a specific use case**
>
> As our work is an applied study, our primary goal is to develop effective methods that can work well and be deployable to real-world systems. However, we also want to point out that our proposed method is not simply reusing the existing methods like TSRL [1], but introduces a series of refinements. Specifically,
> - We proposed a new T-symmetry enforced thermal dynamics model (TTDM) with specifically designed GNN architecture to capture the domain knowledge of the DC environment. We only adopted the T-symmetry regularization design from the T-Symmetry Enforced Dynamics Model (TDM) in TSRL to enhance small-sample performance, but the overall model structure is very different.
> - The policy learning part used in our method (Eq. 9) is also different from TSRL. If the reviewer checks the TSRL paper, their policy regularization is actually performed in the latent action space (i.e., regularizing $\|z_{a^{\pi}}-z_a\|^2$). However, we find that this scheme is unstable and could have a negative impact on real-world complex control tasks. As encoding and decoding errors of latent actions could introduce extra noise during learning, causing performance degeneration. Therefore, we directly regularize the policy-induced actions in the original space (i.e., $\|\pi(s)-a\|^2$), which empirically provides better performance.
> - Moreover, the T-symmetry consistent latent space data augmentation in TSRL is also not used in our method, as we find its benefit is negligible and sometimes could have a negative impact.
> - Lastly, We also designed a safety-aware reward function specifically for the DC cooling system optimization problem.
>
> > **Q1. Apart from the reward function, what are the other measures taken to prevent safety violations?**
>
> When developing our deployed system, we also implemented a rule-based fail-safe mechanism that allows for automatically switching back to the original PID control if the RL policy encounters potential safety issues. For example, if the temperature at a certain location exceeds a pre-defined safety threshold in the server room, the system will automatically switch the nearest ACU to PID control mode. However, throughout our real-world experiments in the commercial data center (1900 hours in total), this safety assurance mechanism has never been triggered during any of our experiments. We think this is another evidence to demonstrate the effectiveness and robustness of our proposed method.
>
> > **Q2. Have the authors tried Offline-Online methods? wherein the policy trained using offline RL deployed is constantly improved using new data obtained?**
>
> - As we are dealing with a mission-critical system (thermal safety violations will cause serious consequences for DC operators), hence it is not possible to conduct policy learning online with the real DC system. Existing offline-to-online RL methods have no guarantee on the quality of learned policies during the online learning stage, directly interacting with the real system during policy learning can be very dangerous. This makes offline RL the only option for such industry control tasks. Moreover, even for our offline learned policies, we typically need to go through an open-loop control inspection (inspecting the policy outputs with real-time data input, but without direct control) by a domain expert to evaluate their reliability, before deploying them into the real system for closed-loop control.
> - Although it is not possible to directly perform offline-to-online learning, as we have discussed in Appendix A Line 725-728, we can periodically re-collect new operational data every few months, and use the new data to fine-tune the previous policies through another round of offline policy learning. We think this is a more viable approach to address the concern of the reviewer on improving policy using new data.

---

> ### Author Response · Authors · 2024-11-20
> **Response to Reviewer JenY (Part 2/2)**
>
> > **Q3. It seems like CQL does perform better than the proposed method in Figure 6, but at the cost of safety. What was the reward function used for CQL?**
>
> In all our experiments conducted on the testbed, all baseline algorithms used the same reward function as Eq.(1). The poor performance of CQL is partly attributed to its poor generalizability and sample efficiency under the small-sample setting, which is also reported in the literature [1, 2]. CQL uses an over-conservative value regularization scheme, which pushes down Q-values for all OOD actions. In small data settings, the majority of the state-action space will become OOD, under this case, CQL will produce a highly distorted value function which makes it impossible to learn a good policy with reasonable OOD generalization capability.
>
>
> [1] Look beneath the surface: Exploiting fundamental symmetry for sample-efficient offline RL. NeurIPS 2023.
>
> [2] When Data Geometry Meets Deep Function: Generalizing Offline Reinforcement Learning. ICLR 2023.

---

> ### Comment · Reviewer_JenY · 2024-11-20
>
> Thank you for your response, I keep my positive score.

---

> > ### Author Response · Authors · 2024-11-22
> > **Thanks for the positive feedback on our work!**
> >
> > Dear reviewer JenY,
> >
> > Thank you for the positive feedback and recognition of our work. We really appreciate the effort you put into the review phase. Thanks again!

---

### Official Review · Reviewer_TQKL · 2024-11-04

**Soundness:** 2
**Presentation:** 3
**Contribution:** 2
**Rating:** 5
**Confidence:** 4

**Summary:**

The authors present a physics-informed offline reinforcement learning framework for optimizing energy efficiency in data center cooling systems, addressing critical challenges like limited data and safety constraints. Using a graph neural network model that respects time-reversal symmetry, the framework enables efficient and robust policy learning from real-world operational data. The authors claimed that this method was successfully deployed in a large-scale commercial data center and achieved 14-18% energy savings over 1300 hours without violating safety constraints. The work demonstrates the potential of offline RL for complex, data-limited industrial applications and calls for a shift from simulation-based benchmarks to real-world problems for more practical and impactful RL research.

**Strengths:**

1. The proposed solution integrates a physics-informed dynamics model to accurately capture the complex thermal behavior within the server room, paired with a graph neural network that embeds domain knowledge to reduce data requirements.
2. The authors claim that this approach produces well-structured and generalizable latent representations, facilitating a sample-efficient offline RL algorithm that maximizes the value function in latent space with appropriate regularization.
3. The implementation includes a safety-aware reward function to ensure operational reliability.
4. The premise of this paper is that as offline RL enables efficient policy learning from pre-collected data, eliminating the risks and costs associated with continuous interaction in safety-critical or resource-constrained environments, it is more effective and practical than online RL.

**Weaknesses:**

1. Claimed but Not Established: The paper asserts strong out-of-distribution (OOD) generalization capabilities and effectiveness with limited real-world data, but these claims are insufficiently substantiated.
2. Lack of Industry Baseline: The real-world validation experiments show 14-18% energy savings in DC cooling without safety violations, but the paper is unable to present any well-defined industry practice baseline with similar objectives for comparison under similar constraints. The comparison lacks fairness. Also, a thorough optimization metric for a data center operation should factor in elements other than safety violations.
3. Model Generalizability & Insufficient Benchmark Comparison: The method heavily relies on modeling, but the generalizability and robustness of the modeling technique remain unverified. The method's performance is not evaluated against established benchmarks, limiting the validation of its general effectiveness.
4. Data and Experiment Limitations: The method's performance evaluation is constrained by the definitions of the experimental setup and the data distributions used in this study, which is not standardized.
5. Minimal Algorithmic Novelty: The approach offers little innovation compared to existing methods, limiting its algorithmic contribution.
6. Comparison with other Physics-informed modeling: The paper does not convincingly demonstrate how the proposed approach is superior to well-established physics simulation models bootstrapped with collected data that use online RL to train. The claimed higher sample efficiency with the physics-informed model is not unique, as similar benefits are observed with both online and offline RL approaches.
7. Unclear Baseline Performance: There is an inadequate explanation for why aggressive baseline methods like CCA and CQL achieve lower energy consumption but fail to maintain critical thermal safety.

In summary, this paper would be better suited for a domain conference centered around physics modeling and the application of standard AI techniques for optimization.

**Questions:**

Please refer to the weaknesses.

---

> ### Author Response · Authors · 2024-11-20
> **Response to Reviewer TQKL (Part 1/3)**
>
> We thank the reviewer for the comments. Regarding the concerns of the reviewer, we provide the following responses.
>
> > **W1. The paper asserts strong out-of-distribution (OOD) generalization capabilities and effectiveness with limited real-world data, but these claims are insufficiently substantiated.**
>
> - In our DC cooling system optimization problems, there are about 100 states and 20-22 continuous actions, but we can only collect 140k-180k historical operational data (about 15~18 months, see detailed description in Appendix B). The data size is quite small compared to the scale of the control optimization problem. By comparison, typical offline RL benchmarks like D4RL often use 1 million data samples to learn simple tasks like MuJoCo (typically fewer than 30 states and actions combined). The complexity of our problem is far beyond the D4RL tasks but with much smaller datasets. Under such a small dataset setting, most of the state-action space actually becomes OOD.
> - In Figures 13 and 14 of Appendix F, we plot all the state and action feature distributions in our collected historical dataset of Server Room A in the real-world DC. Due to the use of PID group control for ACUs throughout the historical operation, and infrequent adjustments to the PID-related temperature setpoints, the action patterns of the ACUs system (fan speed and water valve opening) are narrowly distributed. Additionally, the distributions of other state features are mostly concentrated with a single peak, leaving a large proportion of the state-action space uncovered. All these factors pose significant challenges to offline RL policy learning, requiring the model to have strong generalization capability to achieve good performance.
>
>
> > **W2a. Lack of Industry Baseline**
>
> - We'd like to remind the reviewer that until today, PID and MPC controllers are still the default approaches adopted in the DC industry. Although there are some online RL-based studies that claim to solve the DC's cooling system optimization problem, as we have discussed in the introduction and related work, all of them are only trained and tested in over-idealistic simulation environments, **none of them are able to successfully deploy and validate their effectiveness in real DCs**. As real-world DCs are mission-critical systems, it is simply impossible and not allowed to train online RL policies with direct interaction with the real DC environment.
> - The only study in the literature we found that has been successfully deployed and validated in a real DC environment is a data-driven MPC method [1] from Google Research. In our paper, we have specifically implemented this method and used it as a baseline in our testbed evaluation (denoted as "MPC", see Section 4.2).
> - Moreover, in both our real-world production DC experiments and our real-world testbed experiments (see Section 4.1 and 4.2), we compare with the default PID controller, which is the primary control approach of ACUs adopted by the DC industry worldwide. We are not sure what the reviewer meant by the lack of industry baselines. If the reviewer can actually name a concrete deployable industry practice baseline, we are happy to seriously compare it in our final paper.
>
> [1] Lazic, et, al. Data center cooling using model-predictive control. NeurIPS 2018.
>
> > **W2b. Also, a thorough optimization metric for a data center operation should factor in elements other than safety violations**
>
> In our paper, we have not only evaluated safety violations, but more importantly, provided comprehensive results on energy-saving improvement through Air-side Cooling Load Factor (ACLF), a commonly adopted energy-saving evaluation metric in the DC industry. Moreover, in our control quality analysis in Section 4.1 and Figure 3, 4 we also provide results on the improvement of hot and cold aisle temperature patterns after using our control method. In Appendix C.1 and Figure 11, we further provided a performance analysis of our method under drastic server load fluctuations. We hope the reviewer can at least carefully read our paper before making judgments.

---

> ### Author Response · Authors · 2024-11-20
> **Response to Reviewer TQKL (Part 2/3)**
>
> > **W3. The method heavily relies on modeling, but the generalizability and robustness of the modeling technique remain unverified. The method's performance is not evaluated against established benchmarks, limiting the validation of its general effectiveness.**
>
> > **W4. The method's performance evaluation is constrained by the definitions of the experimental setup and the data distributions used in this study, which is not standardized.**
>
> - First, we have conducted over a total of **1900 hours** of real-world experiments in two server rooms of a large-scale commercial DC environment, as well as comprehensive comparative experiments on a small-scale real-world DC testbed environment. All these efforts are made exactly to justify the generalizability (different server rooms, production DC vs. testbed environment) and robustness (extremely long period of real-world testing) of our modeling approach. We are not aware of any other offline RL study that has seriously conducted over a thousand hours of real-world testing.
> - Second, we'd like to remind the reviewer that our paper is applied research rather than developing a general-purpose algorithm. All of our model designs (GNN architecture, physics-informed design, T-symmetry enforcement, safety-aware reward function, etc.) are made to best solve the real-world DC cooling system optimization problem, and build a deployable working system. None of the existing offline RL benchmarks contain tasks similar to our problem. Moreover, our DC cooling system optimization problem contains more than 100 states and 20-22 continuous actions, but given only 140k-180k historical operational data, which is already far more challenging than the widely used offline RL benchmark like D4RL (using 1 million data samples to learn simple tasks).
>
> > **W5. The approach offers little innovation compared to existing methods, limiting its algorithmic contribution.**
>
> As we have discussed previously, our work is applied research, our primary goal is to develop effective methods that can work well and be deployable to real-world systems. However, we also want to point out that our proposed method is not simply reusing the existing methods like TSRL [2], but introduces a series of refinements. Specifically,
> - We proposed a new T-symmetry enforced thermal dynamics model (TTDM) with specifically designed GNN architecture to capture the domain knowledge of the DC environment. We only adopted the T-symmetry regularization design from the T-Symmetry Enforced Dynamics Model (TDM) in TSRL to enhance small-sample performance, but the overall model structure is very different.
> - The policy learning part used in our method (Eq. 9) is also different from TSRL. If the reviewer checks the TSRL paper, their policy regularization is actually performed in the latent action space (i.e., regularizing $\|z_{a^{\pi}}-z_a\|^2$). However, we find that this scheme is unstable and could have a negative impact on real-world complex control tasks, as encoding and decoding errors of latent actions could introduce extra noise during learning, causing performance degeneration. Therefore, we directly regularize the policy-induced actions in the original space (i.e., $\|\pi(s)-a\|^2$), which empirically provides better performance.
> - Moreover, the T-symmetry consistent latent space data augmentation in TSRL is also not used in our method, as we find its benefit is negligible and sometimes could have a negative impact.
> - Lastly, We also designed a safety-aware reward function specifically for the DC cooling system optimization problem.
>
> [2] Cheng et al. Look beneath the surface: Exploiting fundamental symmetry for sample-efficient offline RL. NeurIPS 2023.

---

> ### Author Response · Authors · 2024-11-20
> **Response to Reviewer TQKL (Part 3/3)**
>
> > **W6. The paper does not convincingly demonstrate how the proposed approach is superior to well-established physics simulation models bootstrapped with collected data that use online RL to train. The claimed higher sample efficiency with the physics-informed model is not unique, as similar benefits are observed with both online and offline RL approaches.**
>
> - First, the "physics-informed modeling" in our paper is completely different from the "physics simulation models" referred to by the reviewer. Our "physics-informed modeling" refers to embedding the domain knowledge of spatial and control dependencies among sensors and ACUs, as well as embedding the generic T-symmetry in our model design. Whereas the "physics simulation models" are the physics or data-driven models that model or simulate the dynamcal processes of the system. Note these two notions are completely different.
> - Second, our proposed TTDM is primarily used to provide well-behaved and generalizable representations to facilitate downstream offline policy learning, rather than used as a dynamics or simulation model to generate new synthetic data as in typical model-based RL methods. In other words, we never used the TTDM to generate new data or bootstrapped them with the collected offline data for policy learning. Under the small dataset setting, learning a well-behaved representation is more reliable for offline policy learning as compared to using model-generated data, as the dynamics model can be poorly learned with limited samples (we refer the reviewer to check [2] for more detailed discussions).
>
>
> > **W7. Unclear Baseline Performance: There is an inadequate explanation for why aggressive baseline methods like CCA and CQL achieve lower energy consumption but fail to maintain critical thermal safety.**
>
> - CCA [3] is based on the off-policy RL algorithm DDPG, which essentially follows an online RL design without considering offline data regularization, and thus can perform poorly under a pure offline RL setup, especially under small-sample settings.
> - The poor performance of CQL in the testbed experiments is mostly attributed to its poor generalizability and sample efficiency under the small-sample setting, which is also reported in the literature [2, 4]. CQL uses an over-conservative value regularization scheme, which pushes down Q-values for all OOD actions. In small data settings, the majority of the state-action space will become OOD, under this case, CQL will produce a highly distorted value function and negatively impact the generalization performance during policy learning.
>
> [3] Li, et al. Transforming cooling optimization for green data center via deep reinforcement learning. IEEE transactions on cybernetics.
>
> [4] Li, et al. When Data Geometry Meets Deep Function: Generalizing Offline Reinforcement Learning. ICLR 2023.

---

> > ### Comment · Reviewer_TQKL · 2024-11-21
> > **Thank you for your response. Unfortunately these responses are not adequate to change my original rating.**
> >
> > I will keep my original rating for this paper as "reject" based on several concerns in my earlier review. The rebuttal response was unable to address these gaps effectively,

---

### Author Response · Authors · 2024-11-20
**General Response and Revision Summary**

We thank all the reviewers for the detailed and constructive comments, especially Reviewer 4H4V and JenY for their positive feedback on our paper. In the following, we report the additional real-world experiment results and revision summaries of our paper.

### **Additional real-world experiments on commercial DC**

In the past two months, we have negotiated with the commercial DC operator and successfully gained their permission to fully control all ACUs in the server room. We have conducted additional real-world experiments in the production DC environment to control 6 ACUs and, subsequently, all ACUs (10 or 11) in the server room. The following tables report the experimental results for Server Rooms A and B, respectively. Compared to controlling 4 ACUs as reported in our original submission, the maximum improvement in energy saving has increased to **21%** when all ACUs are controlled. Our model has now been operated effectively and safely for over **1900 hours** in the real-world commercial DC. We have also included these new experimental results in Table 1 of our revised paper.

| Server Room A | PID        | Ours (Control 6 ACUs) | Ours (Control all ACUs) |
| -------- | ---|-------- | -------- |
|   |  May 6th - 09:50 - 17:20 | Sep 23 11:00 - Sep 29 10:30 | Nov 11 16:30 - Nov 12 16:30 |
| Server average electric power (kW) | 552.17 | 572.77 | 577.63 |
| Server energy consumption (kWh) | 4141.34 | 82199.92 | 13864.55 |
| ACU average electric power (kW) | 23.82 | 20.78 | 19.66 |
| ACU energy consumption (kWh) | 178.73 | 2981.84 | 471.8 |
| ACLF (%) | 4.32 | 3.63 (↓16%) | 3.40 (↓21%) |

| Server Room B | PID        | Ours (Control 6 ACUs) | Ours (Control all ACUs) |
| -------- | ---|-------- | -------- |
|   |  May 6th - 09:50 - 17:20 | Sep 23 11:00 - Sep 29 10:30 | Oct 30 10:10 -  Nov 1 17:30 |
| Server average electric power (kW) | 602.04 | 576.52 | 619.55 |
| Server energy consumption (kWh) |  4520.42 | 82746.24 | 34302.42 |
| ACU average electric power (kW) | 36.38 | 29.15 | 30.06 |
| ACU energy consumption (kWh) |  272.9 | 4183.44 | 1663.22 |
| ACLF (%) | 6.04 | 5.06 (↓16%) | 4.85 (↓20%) |

---
### **We have revised our paper (highlighted in blue text color). The modifications are summarized as follows:**

- (For all reviewers) We added the new results of controlling 6 and all ACUs in the commercial DC's two server rooms. The new results are updated in Tabel 1 as well as discussed in multiple places in the main text.
- (For reviewer 4H4V) We revised the caption of Table 1 to add an additional explanation of ACLF.
- (For reviewer FhGh) In Figure 5, we added additional results with our model controlling 6 and all 10/11 ACUs in the server room to further demonstrate the positive energy-saving impact of controlling more ACUs.
- (For reviewer 4H4V) We conducted additional experiments to analyze the impacts of our method on the upstream water-side cooling system. The results are reported in Appendix C.3 in our revised paper.
- (For reviewer FhGh) We added the learning curves of our method in Appendix G.

---

### Meta-Review · Area_Chair_8sTP · 2024-12-20

**Metareview:**

Summary:
The authors propose a physics-informed offline RL framework to optimize energy efficiency in data center cooling systems, which addresses the challenges such as limited data availability and strict safety constraints. The framework incorporates a graph neural network (GNN) model designed to respect time-reversal symmetry, and it enables efficient and reliable policy learning from real-world operational data.

Strengths:
The proposed methods represent a significant accomplishment, which have been successfully deployed in a real-world data center. This approach transcends the application of existing algorithms by utilizing state-of-the-art offline RL techniques tailored specifically for the data center cooling problem. The integration of a GNN with a physics-informed loss function looks both innovative and well-founded.

Weakness:
The literature review on real-world RL-based cooling systems and the description of the collected data could be improved, as these aspects were concerned by the reviewers.

Decision:
Although the identified weaknesses exist, they appear straightforward to address. More importantly, the significance and contributions of this work outweigh these shortcomings. I recommend acceptance but strongly encourage the authors to address the mentioned weaknesses.

**Additional Comments On Reviewer Discussion:**

Three out of the four reviewers support the acceptance of this paper. After rebuttal, while the last reviewer is still concerned about the statement of other real-world RL based cooling center and the inconsistency of the description of the collected data, I believe they can be addressed in the revised paper, and I also suggest the authors to do so. Overall, I believe this work has the potential to push the boundary of the relevant research, and it also meets the standard of ICLR.

---

### Decision · Program_Chairs · 2025-01-22

Accept (Poster)